# Diversity of tooth mineralisation patterns at the base of crown chondrichthyans
M. Greif [1] ✉, H. Botella [2], T. M. Scheyer [1] & C. Klug [1]

The highly specialised dentitions of modern sharks enable them to exploit a wide range of food sources. Exceptional fossil preservation of three Late Devonian basal chondrichthyan taxa from the Anti-Atlas, Morocco, provides the unique opportunity to study these dentitions in detail, including their tooth histology, replacement patterns, and mineralisation sequences. Thin sections and CT-data of tooth files reveal a high histological diversity and evidence a noticeable disparity in mineralisation patterns early in chondrichthyan evolution. The presence of similar tooth histology and mineralisation patterns in phylogenetically and chronostratigraphically distant chondrichthyan taxa opposes a phylogenetic signal. Although the pseudoosteodont histotype is considered plesiomorphic, we found a high disparity regarding the arrangement of dental tissues in early chondrichthyans. Tooth size differences indicate slow tooth replacement rates for *Ctenacanthus* and *Maghriboselache*. Smaller differences in *Phoebodus* suggest an elevated rate. Tooth retention in *Maghriboselache* might constitute a precursor for the holocephalan evolution of tooth plates.

Modern sharks (neoselachians) exhibit a broad spectrum of trophic strategies[1]. This is partly due to their specialised and morphologically diverse (interspecifically) dentitions[2]. The earliest ancestors of modern sharks date back to the Palaeozoic, potentially as early as the Ordovician[3]. While certain aspects of their dentitions are well documented, others remain largely unknown, including histological diversity, patterns of tooth mineralisation, and modes of tooth replacement.

Chondrichthyan dentitions are composed of tooth files, which consist of rows of individual teeth arranged either alternately or as single files, aligned along the jaw[4]. To maintain a functional dentition, many neoselachians continually replace their teeth. This polyphyodonty is a characteristic feature for many chondrichthyans and likely plesiomorphic[5]. In some cases, a single individual can produce thousands of teeth throughout its life[6]. Each tooth file contains both functional and replacement teeth. Embryonic teeth, which develop internally from tooth buds formed at the dental lamina, initially emerge in the lingual part of each tooth file, erupt into the mouth, and migrate towards the labial part (jaw margin) until they occupy functional positions. This occurs in a 'conveyor belt'-like manner[7,8]. Teeth remain functional for a certain amount of time, and later are shed (sometimes retained[8]) and/or replaced by the next tooth in the file. As a result, teeth at various ontogenetic stages are present in the jaw at any given time[4,5] allowing for the observation of mineralisation sequences through the histology of individual teeth.

The number of functional teeth, tooth replacement rates and the type of replacement (as individual teeth or groups) vary between species[9]. In neoselachians, tooth replacement rates vary from only a few days to multiple weeks[6,10–12]. In fossil taxa, direct estimation of tooth replacement rates remains challenging; however, indirect evidence can yield valuable insights. In sharks, tooth size generally increases with overall body growth, meaning successive teeth within the same tooth file are progressively larger from old to young, thereby producing a size gradient along the file[7,11]. The difference in size between teeth of the same file can be related to the rate of replacement[8,11]. Based on this, several authors have proposed that early chondrichthyans have exhibited comparatively slow tooth replacement rates[6,8,13]. This is inferred from Palaeozoic chondrichthyan fossil material in which tooth files display pronounced size differences, with younger teeth often substantially larger than the functional ones—indicating significant growth prior to their replacement.

Neoselachian shark teeth possess a highly mineralised, triple-layered enameloid cap[14–16] and are classified into three distinct histotypes, defined by the spatial arrangement of the tooth-forming tissues[17,18]. In terms of mineralisation patterns, substantial interspecific variation exists, particularly in the timing and extent of mineralisation of the tooth base and the onset of trabecular dentine mineralisation (often referred to as osteodentine in the literature; see discussion – histotype scheme) between different taxa[5,9,18,19]. While in some taxa mineralisation starts solely with the cusp, in others the tooth base starts to mineralise simultaneously. Enameloid mineralises first, followed shortly by the onset of orthodentine mineralisation. As mineralisation proceeds, the tooth gradually advances anteriorly within the file[5,9,18]. The mineralisation of trabecular dentine typically occurs

[1]University of Zurich, Department of Paleontology, Zurich, Switzerland. [2]University of Valencia, Cavanilles Institute of Biodiversity and Evolutionary Biology, Paterna, Spain. ✉e-mail: merle.greif@pim.uzh.ch

Fig. 1 | Three-dimensional reconstructions of the preserved skeletal parts of *Ctenacanthus concinnus*, *Phoebodus saidselachus* and *Maghriboselache mohamezanei* with emphasis on their tooth files. A ESEFB-LTM-201, *Ctenacanthus concinnus*: overview of the entire specimen, including two tooth files, parts of the jaw cartilage and further teeth. B ESEFB-LTM-203, *Phoebodus saidselachus*: overview of the frontal part of the skull of the specimen and a close-up of the left palatoquadrate and a tooth file. C ESEFB-LTM-202, *Maghriboselache mohamezanei*: overview of the complete jaws and teeth of the specimen and close-up of one tooth file of the frontal left Meckel's cartilage. CT-data and segmentation: Files 5–10[31].

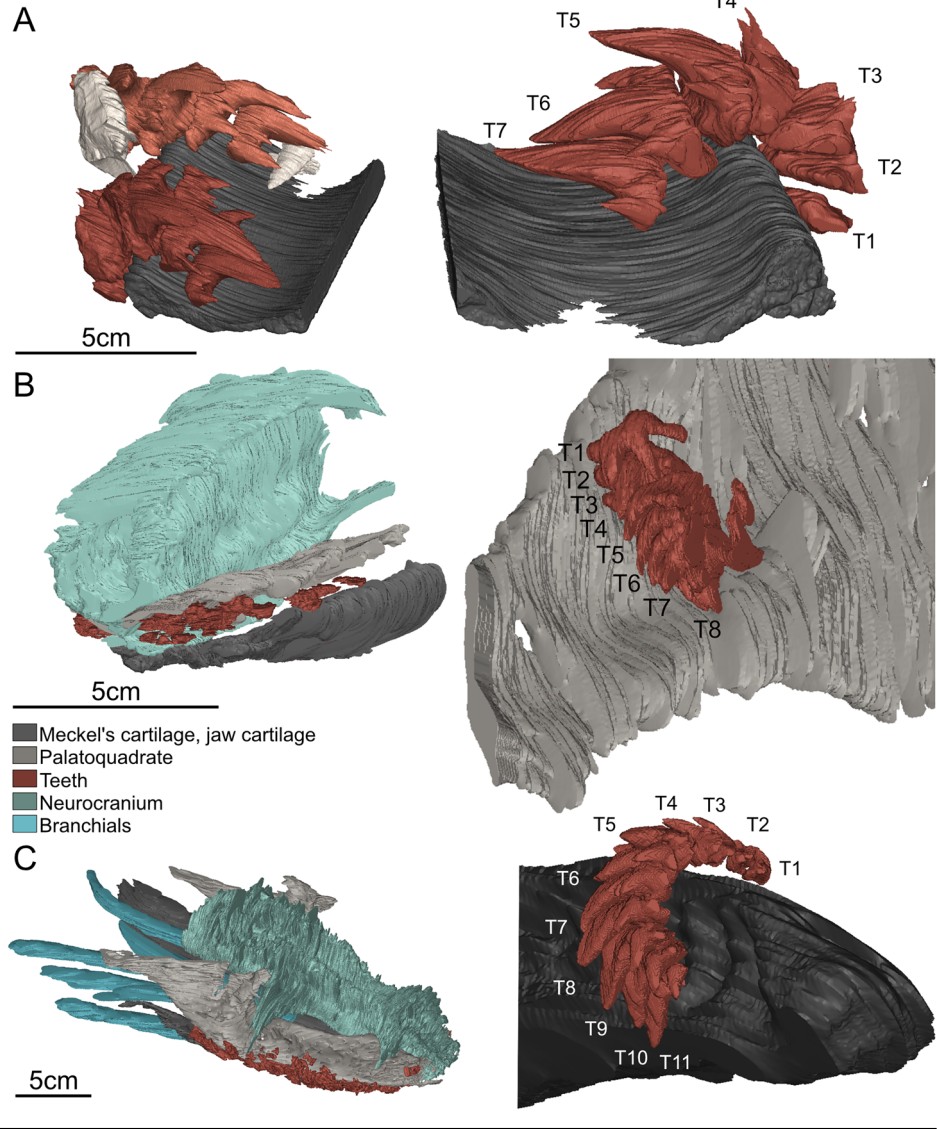

last and begins abruptly. During subsequent development, these tissues become increasingly dense until the tooth is fully mineralised[9].

In early chondrichthyans, overall tooth histology is fairly well studied[15,20–25]. Several studies have examined tooth histology of stem chondrichthyans[21], stem elasmobranchs[21,26] and stem holocephalans[15]. In general, enameloid structures are comparatively simple, consisting of a single layer of crystallite enameloid (SCE)[15,21,24,25,27,28]. An exception is observed in xenacanths (stem elasmobranchs), which lack enameloid[21,26]. However, the relationship between histological patterns and phylogenetic position remains ambiguous. In contrast to the histology itself, the processes of mineralisation in early chondrichthyan teeth have not been studied yet. Regarding early chondrichthyans, the overall tooth histology is comparatively well studied[15,20–25]. However, to study mineralisation processes and the development of teeth within one tooth file requires exceptional fossil preservation. Such a preservation – where teeth are found in articulation within their files – is rarely seen in the fossil record. A select few specimens of the three chondrichthyan species *Ctenacanthus concinnus*, *Phoebodus saidselachus*[29] and *Maghriboselache mohamezanei*[30] from the Late Devonian of Morocco exhibit such exceptional preservation. All three taxa belong to the chondrichthyan crown group (see Klug et al.[30], fig.13): *Ct. concinnus* and *P. saidselachus* represent distantly related stem elasmobranchs, while *M. mohamezanei* is a stem holocephalan (see Klug et al.[30], fig. 13). As such, these taxa represent distant members of the two subclasses, Elasmobranchii and Holocephalii, which constitute the chondrichthyan class.

The new material provides insights into the histology and mineralisation patterns of early chondrichthyan teeth. Especially the latter has, to our knowledge, not been documented before. In this study, we examine the phylogenetic implications of the new data, focusing on the role of tooth mineralisation patterns and replacement rates in the evolutionary and ecological success of cartilaginous fishes. Lastly, we discuss estimates of tooth replacement rates in early chondrichthyans and compare them to modern analogues.

## Results
### Histology
***Ctenacanthus concinnus.*** The teeth of the ctenacanthid *Ctenacanthus concinnus*[23] have a main cusp and two to three cusplets on each side, with the outermost being the largest. In specimen ESEFB-LTM-201, one tooth file includes 7 teeth (T1–T7), which is also represented in the CT data (Fig. 1A). However, the tooth file is incomplete, lacking parts of and/ or possibly complete teeth labially. Numerous small oral denticles (or mucous denticles sensu Peyer[28]) are scattered between the teeth (Fig. 2A; Supplementary Note/Fig. 1, File 1[31]). The maximum crown height is approximately 24 mm in our sections, but larger sizes are known from other Moroccan specimens[32]. The tooth size increases labio-lingually.

A thin monolayer of enameloid caps approximately 4/5th of the main cusps (Fig. 3A). There is no indication of a microstructural differentiation of enameloid at any magnification (Fig. 4F, H). The enameloid layer is

**Fig. 2 | Histological thin sections of the tooth files of *Ctenacanthus concinnus*, *Phoebodus saidselachus* and *Maghriboselache mohamezanei* and schematic drawings. A** ESEFB-LTM-201, *Ctenacanthus concinnus* (photo ID: C_con_001_s1); left: overview photo of the thin section 1; right: schematic drawing showing the jaw cartilage, teeth 7-1 and oral denticles. **B** ESEFB-LTM-203, *Phoebodus saidselachus* (photo ID: Ph_002); left: overview photo of the thin section; right: schematic drawing showing the four tooth families (F1–F4), all preserved teeth within and oral denticles. **C** ESEFB-LTM-202, *Maghriboselache mohamezanei* (photo ID: Mg_004_s1); left: overview photo of the thin section 1 showing Palatoquadrate (PQ) and Meckel's cartilage (MC) and all preserved teeth as well as oral denticles aligned along the jaws. In all drawings, grey gradients represent the mineralisation status from darker colours – an early state of mineralisation, to bright – fully mineralised. For the original high-resolution photos, see File 2–4[31].

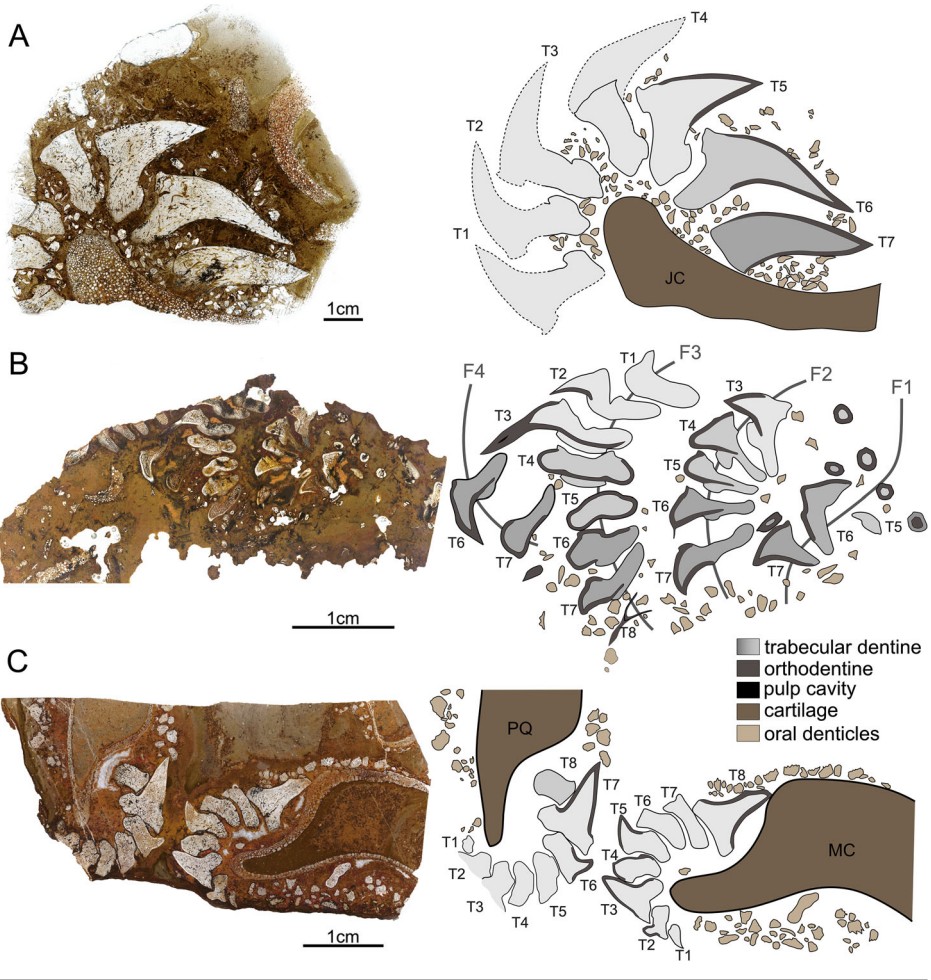

approximately 0.05 mm thick along the labial and lingual sides of the cusp, tapering out towards the tooth base. In the most apical point of the cusp, the thickness is 0.15 mm (Fig. 4F). The hypermineralised enameloid layer covers a homogeneous layer of orthodentine of about 0.5 mm thickness (Fig. 3A), which extends across the entire cusp until the onset of the base (Fig. 3A). The enameloid-orthodentine junction is well defined. The orthodentine contains numerous thin parallel dentine tubules which branch out towards the enameloid junction (Fig. 4H). The inner part of the cusp, as well as the base, is composed of trabecular dentine (Figs. 2A, 4A, B, E, G). The trabecular dentine is porous, and dentinal osteons are irregularly disposed in a twisting, branching network. The difference in dentinal patterns is especially apparent at the orthodentine-trabecular dentine junction. The junction is very clear throughout the entire tooth cusp when the regular pattern of orthodentine meets the irregular trabecular dentine (Fig. 3). In fully developed teeth, trabecular dentine is arranged in a circular pattern with dentine growing around the osteons (Fig. 4A, B). In the apical parts of the tooth, in contrast, the trabecular dentine tubules are more directed and elongated (Fig. 4G). The base of the teeth contains several internal nutrient canals, which are mainly found in the lingual basal area below the button (Figs. 2 and 3A). A thin layer on the outside of the button, as well as the basolabial projection, appears darker with large collagen crystals present (Fig. 4C, D).

***Phoebodus saidselachus***. The teeth of the phoebodontid *P. saidselachus*[29] are tricuspid with small cusplets between the main cusps. The main cusps measure up to 6–7 mm (measured in CT scan, File 7 and 8[31]). The thin section of ESEFB-LTM-203 includes teeth from at least four different tooth families at different developmental stages that contain up to 8 teeth (T1–T8, Figs. 1B, 2B) as well as oral denticles

scattered around the teeth (Supplementary Fig. 1). The labial-most region is lost, meaning the total number of teeth per file in living specimens may have been higher than what is preserved. Unfortunately, the contrast in the CT scan is not high enough for a very precise segmentation. However, the 3D reconstruction confirms 8 teeth per file (Fig. 1B).

Tooth cusps are capped with a thin homogeneous monolayer of enameloid without any indication of microstructural differentiation. Orthodentine fills the cusps, enclosing a small central pulp cavity, and extends both lingually and labially below the crown/base boundary into the peripheral zone of the base, although it does not reach the basal surface. The thickness of the orthodentine is relatively homogeneous across different parts of the tooth, reaching up to 0.5 mm in some specimens. The dentine tubules in the orthodentine are numerous, thin and distinctly branched, particularly towards the outer surface of the tooth. The enameloid-orthodentine junction is well defined, with many dentine tubules crossing the junction and extending into the capping tissue (Fig. 5A). Trabecular dentine occupies the base, including the oral button, where it overlays the orthodentine (Fig. 5G, H). It also fills the lower part of the cusp core. Generally, trabecular dentine is highly porous with large tubules and irregularly arranged dentinal osteons (Fig. 5A, B, F). Several internal nutrient canals (distinguished from trabecular spaces) are identifiable in the base, some of which cross the orthodentine layer, particularly near the oral button, and open on the basal and lingual surfaces of the teeth. No large or primary canals are discernible in the teeth, at least based on the orientation of the sections. The boundary between orthodentine and trabecular dentine is sharply defined along the base and much of the cusp but tapers toward the apex, where orthodentine projections interdigitate with trabecular dentine (Figs. 3C, 5A).

**Fig. 3 | Histological structures in single fully mineralised teeth of *Ctenacanthus concinnus*, *Phoebodus saidselachus* and *Maghriboselache mohamezanei*. A** ESEFB-LTM-201, *Ctenacanthus concinnus* drawing and photo of tooth T5 in thin section 1 (photo ID: C_con_019_s1_T5). **B** ESEFB-LTM-202, *Maghriboselache mohamezanei* drawing and photo of tooth T6 in thin section 2 (photo ID: Mg_039_s2_PQ_T765). **C** ESEFB-LTM-203, *Phoebodus saidselachus* drawing and photo of tooth T3 in tooth file 3 (photo ID: Ph_006_F3T3). The drawings elucidate the boundaries of enameloid, orthodentine and trabecular dentine. For the original high-resolution photos, see File 2–4[31].

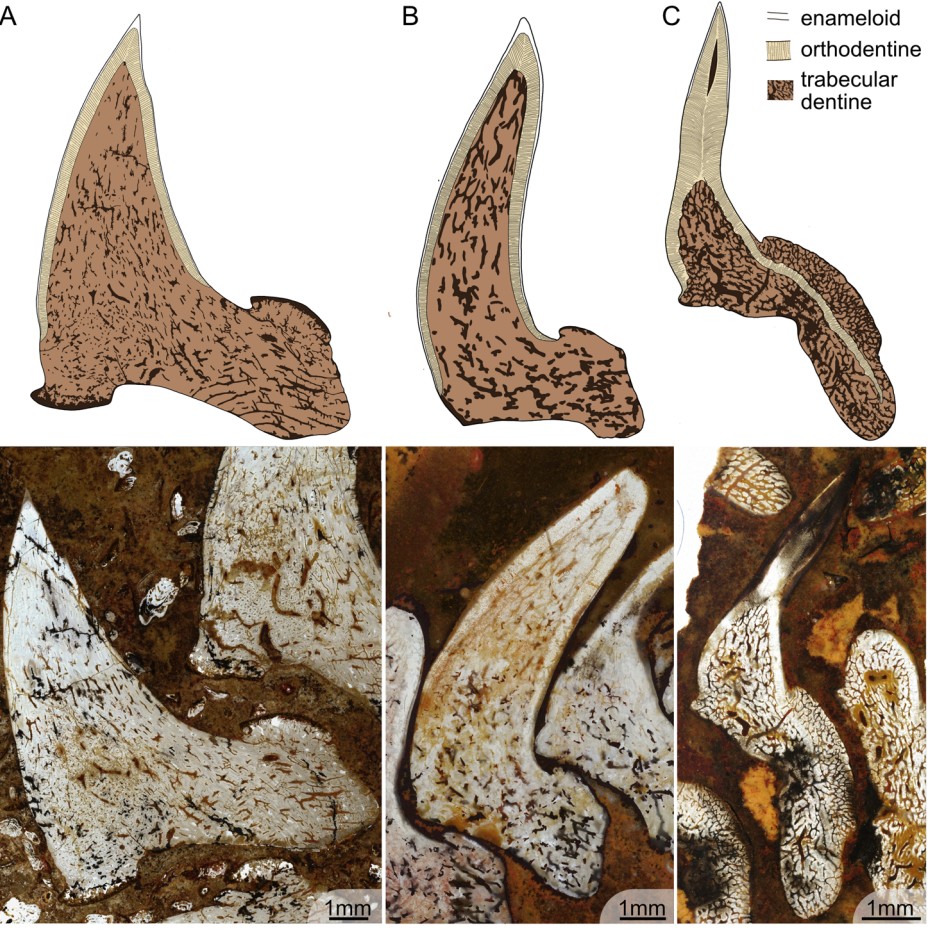

*Maghriboselache mohamezanei*. The teeth of the cladoselachian *M. mohamezanei*[30] have a main cusp and up to three cusplets on each side; the outermost of which is the largest. One tooth file holds 11 teeth as revealed by the CT data (Fig. 1C, Files 9 and 10[31]). Some files, including the ones shown on the thin sections, bear fewer teeth due to preservation. In the specimen ESEFB_LTM-202 studied here, the maximum tooth height is approximately 10 mm. The thin sections show two tooth files in situ, arranged around the palatoquadrate and Meckel's cartilage, as well as numerous oral denticles aligned along the jaw cartilages (Fig. 2C; Supplementary Fig. 1).

The two thin sections of *M. mohamezanei* include two opposing complete tooth files, each including eight teeth at different developmental stages (Fig. 2C). The overall tooth histology is taphonomically altered. However, a homogeneous layer of enameloid is visible in several areas of the teeth (Fig. 6F) but wedges out in many places due to poor preservation. In Section 1, PQT7 and in Section 2, PQT6 (Fig. 2B, File 4[31]), enameloid appears to cover the cusp lingually down to the transition from cusp to root and labially a little further down, not reaching the base. Where present, the enameloid-orthodentine junction is very sharp. A thick layer of orthodentine covers about 80% of the cusp, as in *Ct. concinnus*. Orthodentine tubules are preserved in some areas of the teeth, running perpendicular to the enameloid-orthodentine junction and are thin and poorly branched. (Fig. 6C–E). The orthodentine-trabecular dentine junction is distinct (Fig. 6F). The base is tubule-rich, while trabecular dentine in the apex appears denser (Fig. 3B).

### Sequence of mineralisation during tooth development

Thin sections of all three taxa include seven to eight teeth of individual tooth files, each representing different developmental stages. This allows for an accurate interpretation of the sequence of appearance and maturation of mineralised tissues during tooth development. Mineralisation begins with the outer enameloid layer, followed by the deposition of orthodentine, with trabecular dentine mineralisation occurring last. The preservation of tooth files in situ allows us to follow the overall progression of mineralisation (Fig. 2). However, the thin sections only present a snapshot into development, and not all minor steps are preserved.

In *Ct. concinnus*, the earliest stage of mineralisation is preserved in Section 2, T7 (File 2, Section 2[31]), where a thin layer of dense enameloid and orthodentine is present, along with poorly mineralised trabecular dentine (Fig. 4E). In *P. saidselachus*, the earliest stage of mineralisation is represented by F3T8, which shows a densely mineralised enameloid cap and a thin layer of orthodentine (Fig. 5 H, I, J). In contrast, the youngest preserved teeth (T8) from the palatoquadrate file of *M. mohamezanei* (File 4, Section 2[31]) exhibits a more advanced degree of mineralisation. At this early stage of mineralisation, the cusp is located close to the jaw cartilage, as apparent in the thin section of *Ct. concinnus* (Fig. 2A). As mineralisation progresses and due to the mineralisation of the tooth base, the tooth gradually shifts further away from the jaw.

Subsequent stages of maturation in *Ct. concinnus* and *P. saidselachus* show the presence of enameloid, orthodentine and porous trabecular dentine, as the tooth continued to grow (Fig. 2A, B). In all three taxa, mineralisation progressed in a linguo-labial direction. Beneath the enameloid layer, orthodentine developed rapidly, preceding the onset of trabecular dentine mineralisation (we assume that for the orthodentine to mineralise rapidly, as it shows already in the youngest teeth as a thin layer; however, the term rapid is relative in this context and cannot be quantified). The orthodentine grows centripetally, starting from the apical part of the cusp toward the base, which is not fully formed at this stage. Mineralisation of the trabecular dentine begins simultaneously in the cusp and base, but remains poorly mineralised. This step is apparent in both *Ct. concinnus* and *P.*

**Fig. 4 | Detailed histology of the teeth of ESEFB-LTM-201, *Ctenacanthus concinnus*. A** Trabecular dentine in the base of T6, section 3 (photo ID: C_con_051_s3_T6), showing concentric growth. **B** Trabecular dentine in the base of T6, section 1 (photo ID: C_con_024_s1_T6) showing concentric growth. **C** Dark layer on the button of tooth T5, section 1 (photo ID: C_con_020_s1_T5). **D** Close-up of C, showing *Sharpey's fibres* (photo ID: C_con_021_s1_T5). **E** Early stage of mineralisation in T7, section 2 (photo ID: C_con_031_s2_T7) showing enameloid and orthodentine as well as tubule-rich trabecular dentine. **F** Apex of T7 in section 3 (photo ID: C_con_054_s3_T7) showing enameloid, orthodentine and trabecular dentine. **G** Apical part of T6, section 1 (photo ID: C_con_026_s1_T6) showing elongated trabecular dentine tubules. **H** Close-up of enameloid and trabecular dentine in T7, section 2 (photo ID: C_con_066_s3_T7). ena enameloid, orth orthodentine, trd trabecular dentine. For original high-resolution photos, see File 2[31].

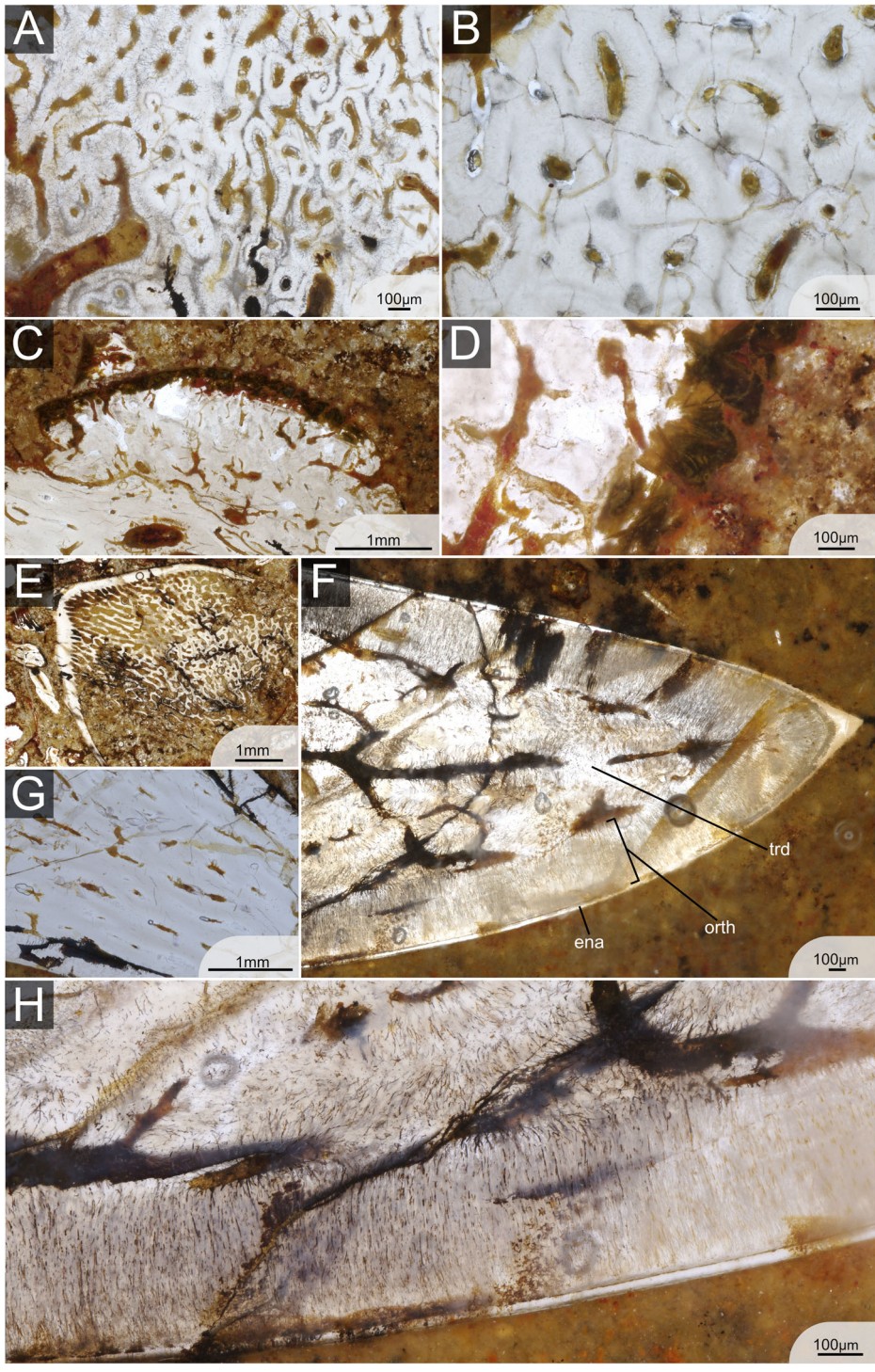

*saidselachus* thin sections, but the trabecular dentine appears denser in *Ct. concinnus* (Fig. 2A, B, T7; note that the orientation of the sections of *P. saidselachus* cuts only the lower part of the cusp and base). In the thin section of *Ct. concinnus*, trabecular dentine is densely mineralised at the apex and tapers out towards the base, implying a comparatively rapid growth of trabecular dentine in this direction (Fig. 2A, T7). Furthermore, mineralisation of trabecular dentine limits the inward growth of orthodentine. The orthodentine-trabecular dentine boundary is well defined in both taxa. In *M. mohamezanei*, taphonomic alteration partly overprints this boundary. While orthodentine is only present in the cusps of *Ct. concinnus* and *M. mohamezanei* (Fig. 2A, C), it extends peripherally into the base after

reaching the maximum thickness, protruding labially and lingually (beyond the position of the oral button) in F1T6, F2T6, and F4T6 of *P. saidselachus* (Fig. 2B). Trabecular dentine densifies gradually from the youngest to the oldest tooth.

Mineralisation of the base is finished at a late stage, apparent in T5 in *Ct. concinnus* and in T4 in *P. saidselachus*. However, as mentioned previously, in *M. mohamezanei* this is achieved already in T8 or T7 in (Fig. 2). T5 of *Ct. concinnus* represents a fully developed tooth with orthodentine having reached its maximum thickness and extending down throughout the entire cusp. Trabecular dentine has mineralised in both the cusp and the labial side of the base. The most lingual part of the base remains less

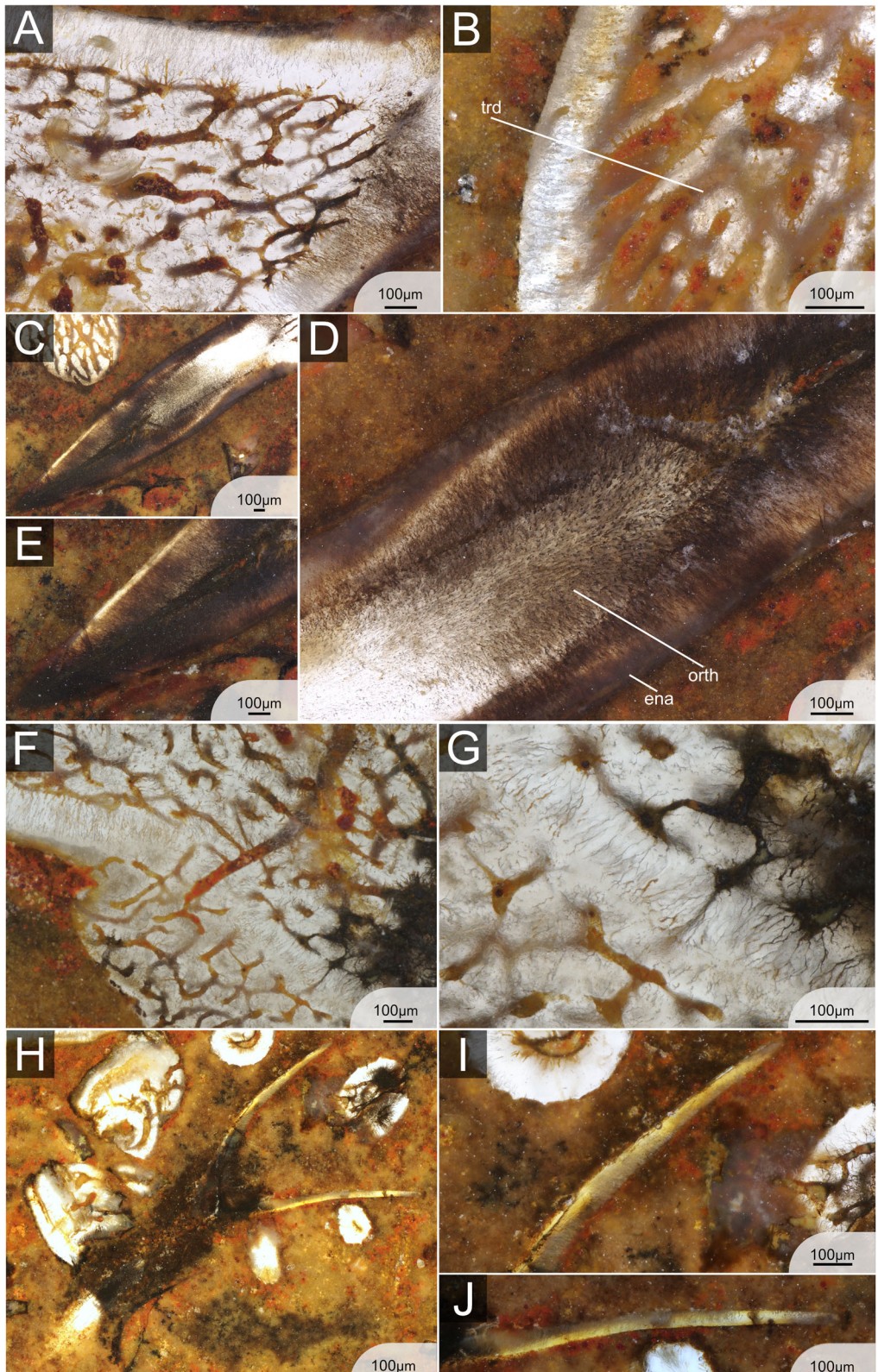

**Fig. 5 | Detailed histology of the teeth of ESEFB-LTM-203, *Phoebodus saidse-lachus*. A** Details of orthodentine and trabecular dentine in tooth 3, file 3 (photo ID: Ph_012_F3T3) showing the junction of orthodentine towards the apex.
**B** Orthodentine and trabecular dentine of tooth 7, file 3 (photo ID: Ph_020_F3T7).
**C** Cusp of tooth 3, file 3 (photo ID: Ph_006_F3T3). Close-up of (**D**) showing the orthodentine structure in the base of the cusp (photo ID: Ph_010_F3T3). **E** Close-up of D showing the enameloid and orthodentine structure in the apical part of the cusp, as well as remnants of the pulp cavity (photo ID: Ph_007_F3T3). **F** Tooth 3, file 3, details of orthodentine in the base and overlaying trabecular dentine of the button (photo ID: Ph_015_F3T3). Close-up of (**G**) (photo ID: Ph_017_F3T3). **I** F3T8 first stage of mineralisation (photo ID: Ph_035_F3T8). **H** Close-up of (**I**), labial side, showing orthodentine and a thin layer of enameloid (photo ID: Ph_037_F3T8). **J** Close-up of (**I**), lingual side, showing a thin orthodentine layer (photo ID: Ph_036_F3T8). ena enameloid, orth orthodentine, trd trabecular dentine. For original high-resolution photos, see File 3[31].

**Fig. 6 | Detailed histology of the teeth of ESEFB-LTM-202, *Maghriboselache mohamezanei*.**
**A** Details of trabecular dentine in tooth 7, palato-quadrate, section 1 (photo ID: Mg_030_s1_PQ_T7).
**B** Details of trabecular dentine in tooth 7, Meckel's cartilage, Section 1 (photo ID: Mg_029_s1_MC_T7). **C** Apex of tooth 3, Meckel's cartilage, section 1 (photo ID: Mg_010_s1_MC_T3) showing enameloid, orthodentine and trabecular dentine. **D** Details of tooth 3, Meckel's cartilage, section 1 (photo ID: Mg_016_s1_MC_T3) showing enameloid and thin parallel dentine tubules of the orthodentine. **E** Close-up of tooth 3, Meckel's cartilage, section 1 (photo ID: Mg_015_s1_MC_T3). **F** Close-up of emameloid and orthodentine of tooth 6, palatoquadrate, Section 2 (photo ID: Mg_042_s2_PQ_T76). ena enameloid, orth orthodentine, trd trabecular dentine. For original high-resolution photos, see File 4[31].

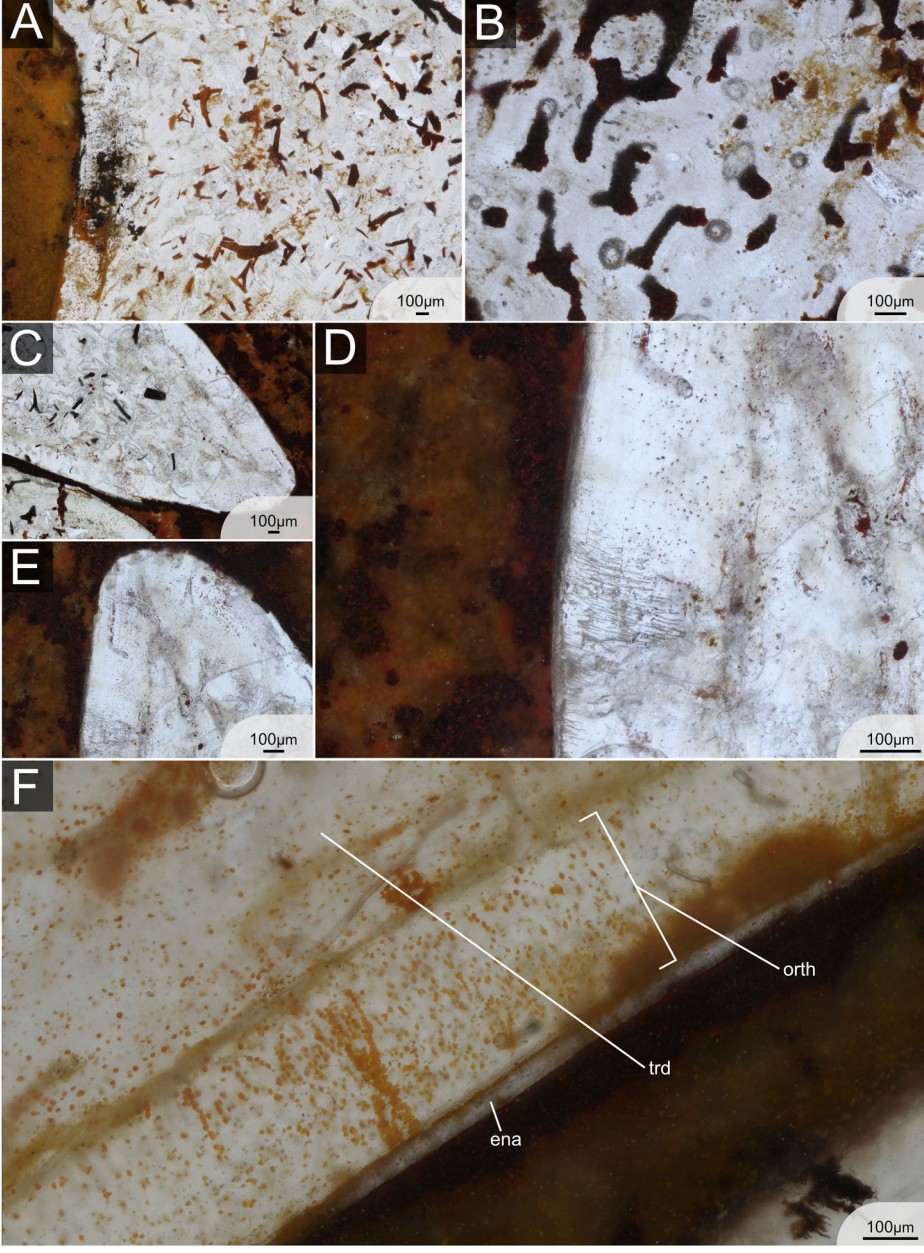

mineralised at this stage. A darker layer on the button as well as on the basolabial projection of the teeth (Fig. 4C, D) indicates the presence of connective tissues (likely Sharpey's fibres[33]) which facilitate consecutive tooth articulation.

Teeth F2T5 and F4T6 of *P. saidselachus* illustrate the sequence of trabecular dentine mineralisation forming the oral button overlying the already formed orthodentine layer. Tooth F3T3 represents a (nearly) fully formed tooth. It shows the complete central cusp with a fully mineralised trabecular dentine core retaining a small central pulp cavity, and the completed mineralisation of trabecular at the basal part of the cusp and root.

## Discussion

The teeth of all three studied taxa contain enameloid, orthodentine and trabecular dentine. However, the arrangement of these tissues differs between taxa and does not appear to be directly linked to their phylogenetic position. While the cladodont teeth of the stem elasmobranch *Ct. concinnus* and the stem holocephalan *M. mohamezanei* exhibit a similar histology, the teeth of the other stem elasmobranch *P. saidselachus* show significant

differences. Notably, ctenacanthiforms and phoebodontiforms are not closely related (Klug et al., fig. 13[30]). Hence, histological differences between the two are not unexpected, particularly considering that tooth histology can vary within members of the same family (Fig. 7).

All three studied taxa show an outer layer of enameloid. This was formerly described in other early chondrichthyans, such as other ctenacanthiforms (e.g., *Ctenacanthus compressus*)[15], or symmoriiforms like *Akmonistion zangerli*[34] and the cladoselachiform *Cladoselache kepleri*[15] (stem holocephalans). Xenacanths are the only stem elasmobranchs that lack enameloid[15,26,35] (Fig. 7). Some other stem chondrichthyans also lack enameloid and show a different histology, such as *Aztecodus*[21]. In *Ct. concinnus* and *M. mohamezanei*, the enameloid covers the entire cusp, while in *P. saidselachus*, the enameloid cover is restricted to the upper part of the cusp (Fig. 3). In other phoebodonts, enameloid is known to cover the outer cusps until far down but not the central one, e.g., *Thrinacodus*, *Harpago*[36], and *Jalodus australiensis*[15]. In all three taxa, the enameloid does not show an indication of a microstructural differentiation at any magnification, suggesting the presence of SCE (single crystallite enameloid)[14]. SCE is generally

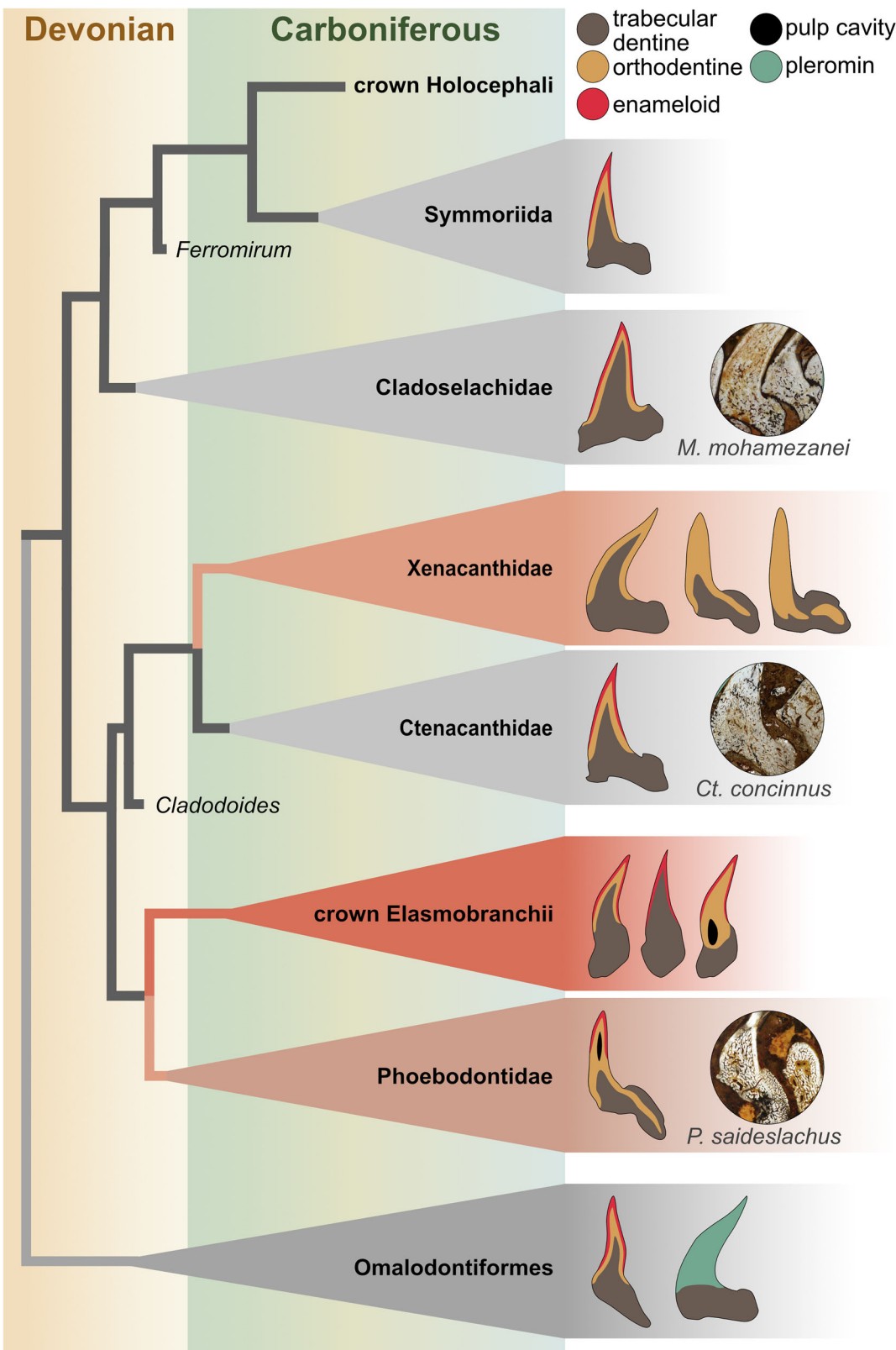

**Fig. 7 | Dental histology and disparity across the chondrichthyan phylogeny.** Symmoriids, cladoselachids and ctenacanthids possess teeth composed of enamel, orthodentine and trabecular dentine (pseudoosteodont). Stem chondrichthyans show histological variety, including pleromin in the crown. Xenacanths are characterised by the lack of enameloid but the composition of orthodentine varies. In some cases (*Reginaselache morrisi*), orthodentine intrudes into the base of the tooth, similar to *Phoebodus* saidselachus, where additionally enameloid and a pulp cavity are preserved. The crown elasmobranchs have teeth of either pseuoosteodont, osteodont or orthodont histotype. The three taxa studied here are shown with their histological pictures. Phylogenetic tree adapted from Klug et al.[31]. Tooth morphologies do not exactly reflect reality. Histological data for Palaeozoic chondrichthyans from the literature as shown in Supplementary Table 1.

accepted to be the ancestral state of enameloid in chondrichthyans[15,22] and is thus expected in these stem-group representatives. However, to exclude differentiation of enameloid as seen in neoselachians (i.e., layered enameloid[14]), further examination using a scanning electron microscope and acidic etching would be needed (as performed by Gillis and Donoghue[15] or Singh et al.[37]).

Besides enameloid, our specimens show further differences in the arrangement of orthodentine and trabecular dentine. *P. saidselachus* possesses orthodentine in the base (Fig. 3), a feature also observed in xenacanths, despite their closer phylogenetic relationship to ctenacanths (Fig. 7). The trabecular dentine in *P. saidselachus* remains relatively porous compared to that in *Ct. concinnus* and *M. mohamezanei*. Trabecular dentine in ctenacanths exhibits dentinal osteons that are irregularly disposed in a twisting, branching network, as also described in further early chondrichthyans like *Hybodus*[27]. Such differences denote a high disparity of tooth histology within early stem-group chondrichthyans, suggesting the absence of a phylogenetic signal. This is also apparent in modern sharks (neoselachians). Jambura et al.[17] found that consistent phylogenetic or functional signals cannot be inferred from the different histotypes. They furthermore point out that the pseudoosteodont type is suggested to be plesiomorphic for all modern sharks, while highlighting that the histotype scheme is reductionist[17].

Classically, two histotypes have been described for elasmobranchs: orthodont and osteodont[18,38]. The definition of these histotypes was recently reviewed, and a third was incorporated: the pseudoosteodont histotype[9,19] (a term initially proposed by Herman et al.[39,40]). Orthodont teeth are formed by orthodentine in the crown and trabecular dentine in the base, and during their formation, the pulp cavity reduces, but remains hollow[18,28]. Osteodont teeth lack a hollow pulp cavity and are entirely filled with trabecular dentine, capped by an outer enameloid layer in the cusps. This histotype is named after the term osteodentine, which is often used as a synonym for trabecular dentine. However, the term osteodentine implies the presence of bone cells, which are not present in the trabecular dentine of chondrichthyans[28]. Teeth of the pseudoosteodont histotype do not have a hollow pulp cavity either, and during mineralisation, trabecular dentine intrudes from the base into the pulp cavity and fills it entirely. The trabecular dentine core is surrounded by orthodentine and capped by enameloid.

The teeth of *Ct. concinnus* and *M. mohamezanei* can be broadly classified as of the pseudoosteodont type. However, the histology of *P. saidselachus* does not conform to any known histotype. This is due to the arrangement of orthodentine, which extends from the crown into the base, resembling the histology observed in the xenacanthids *Reginaselache morrisi*[26] or *Mooreodontus indicus*[35]. In contrast, in all three described histotypes, the base is composed solely of trabecular dentine. Additionally, a small, hollow pulp cavity surrounded by orthodentine and capped by enameloid is present at the apical part of each cusp, as seen in F3T3 (Fig. 2B). Teeth of the pseudoosteodont and osteodont types, by contrast, lack a pulp cavity. Teeth of the osteodont type additionally lack orthodentine in the cusp.

The available data suggests that a pseudoosteodont histology is plesiomorphic in stem holocephalans. However, this conclusion cannot be extended to stem elasmobranchs. The pseudoosteodont histotype is evident in ctenacanths, as well as some stem chondrichthyans and crown elasmobranchs (Fig. 7). However, other groups, such as phoebodonts and xenacanths[26,35] exhibit greater histological diversity (Fig. 7). These examples highlight a considerable histological disparity among early chondrichthyans, which is evident even at a small sample size. We collected histological data of 13 Palaeozoic taxa, encompassing the entire tooth (Supplementary Table 1). Notably, 50% of these taxa are not of the pseudoosteodont type (*sensu* Jambura[9]).

Given the histological diversity and disparity in early chondrichthyans highlighted by *P. saidselachus*, it is evident that the current histotype scheme does not fully reflect the variability of chondrichthyan teeth. This is especially true when considering lineages of the chondrichthyan stem group. Although the separation between these histotypes has been used for more than a century, and is still used (see Moyer et al.[18]; Jambura et al.[17] and references therein), several authors have pointed out that the current histotype concept is too reductionist. In fact, Peyer[28] suggested that there are many intermediate conditions showing varying thicknesses and mineralisation modes of dentine in early chondrichthyans. Moyer et al.[18] mention that the two classical histotypes are extremes, and that the concept of histotypes is a gross morphological one that does not necessarily imply much about the dental tissues themselves. Jambura et al.[17] conclude that histotypes are overused descriptors that can neither infer consistent phylogenetic nor functional signals. Thus, attributing a tooth to a histotype is only useful as a rough initial classification.

Histological sections provide detailed insights into the sequential development and mineralisation of dental tissues during tooth formation in *P. saidselachus* and *Ct. concinnus*. In *M. mohamezanei*, however, such detail is lacking, as even the earliest teeth within the dental file are already almost completely mineralised, preventing a detailed reconstruction of the mineralisation process. Consequently, the following section will focus on the former two taxa.

In *P. saidselachus* and *Ct. concinnus*, tooth mineralisation starts with the formation of an enameloid cap followed by the onset of orthodentine mineralisation beneath it, progressing centripetally around the hollow pulp cavity. The earliest observable stage of mineralisation in *Ct. concinnus* (T7 of Section 2, Fig. 4E; File 2[31]; *Ct. concinnus*, Section 2: C_con_028_s2.tif) closely resembles the second stage of mineralisation in *P. saidselachus* (Fig. 2B, F2T7, F1T7 and F4T7), characterised by the presence of enameloid, orthodentine, and poorly mineralised, spongy trabecular dentine. This similarity suggests that either the initial stages of mineralisation are not preserved in *Ct. concinnus* or that the mineralisation is more rapid in this taxon.

The most important differences concern the mineralisation of orthodentine and trabecular dentine (Table 1). While in *Ct. concinnus*, the orthodentine layer extends only to the periphery of the cusp and terminates there, in *P. saidselachus*, orthodentine extends down into the tooth base. The mineralisation of trabecular dentine also differs between the two taxa in terms of both onset and filling of the tooth. In *Ct. concinnus*, trabecular dentine first mineralises in the apical part of the crown, gradually extends downwards while filling the entire pulp cavity, finishing with the mineralisation of the base. In *P. saidselachus*, trabecular dentine appears almost simultaneously in the pulp cavity of the crown and in the base, continuing to mineralise until the base is fully formed last. The mineralisation of trabecular dentine limits the inward growth of orthodentine. Such differences in the sequence of tooth mineralisation have recently been described in neoselachians[5,9,17,18], though comparative data for early chondrichthyans is not available.

The mineralisation sequence of trabecular dentine in *Ct. concinnus* generally follows a pattern similar to that described for most lamniform shark teeth of the osteodont histotype[9,18]. In particular, the teeth of *Carcharodon carcharias* mineralise in a comparable manner, with trabecular dentine appearing first in the crown and progressively mineralising downwards toward the base (Fig. 1A; *Ct. concinnus* in comparison to Peyer[28], Pl. 17A and Moyer et al.[18], Fig. 3C, D). However, a major difference between *Ct. concinnus* and lamniform teeth of the osteodont histotype is the development of orthodentine in *Ct. concinnus*, which is absent in the latter. In contrast, the mineralisation sequence in *P. saidselachus* is more similar to that found in certain lamniform neoselachians with a pseudoosteodont histotype, such as *Pseudocarcharias kamoharai*[19], where trabecular dentine mineralises simultaneously in both the pulp cavity of the crown and in the base. However, in other neoselachians with pseudoosteodont histotype, like the carcharhiniform *Hemipristis elongata*[9] or the lamniform *Cetorhinus maximus*[19], trabecular dentine mineralises first in the base and extends apically only after the formation of orthodentine below the enameloid in the crown. The mineralisation sequence in *P. saidselachus* is almost identical to that observed in the extant squalomorph shark *Squatina squatina* (Supplementary Note/Fig. 2) where the orthodentine layer extends downwards

**Table 1 | Tooth mineralisation stages during development**

| Tooth | Enameloid | Orthodentine | Trabecular dentine, crown | Trabecular dentine, base |
|---|---|---|---|---|
| *Ctenacanthus concinnus* | | | | |
| T7 | Present, thin | Present, thin | Start of mineralisation | Absent |
| T6 | Fully mineralised | Fully mineralised | Partially mineralised | Present, start of mineralisation |
| T5 | Fully mineralised | Fully mineralised | Mainly mineralised | Mainly mineralised |
| T4 | Fully mineralised | Fully mineralised | Fully mineralised | Mainly mineralised |
| T3 | Fully mineralised | Fully mineralised | Fully mineralised | Fully mineralised |
| T2 | Fully mineralised | Fully mineralised | Fully mineralised | Fully mineralised |
| T1 | Fully mineralised | Fully mineralised | Fully mineralised | Fully mineralised |
| *Maghriboselache mohamezanei* | | | | |
| T8 | Fully mineralised | Fully mineralised | Partially mineralised | Fully mineralised |
| T7 | Fully mineralised | Fully mineralised | Present, fully mineralised | Fully mineralised |
| T6 | Fully mineralised | Fully mineralised | Present, fully mineralised | Fully mineralised |
| T5 | Fully mineralised | Fully mineralised | Present, fully mineralised | Fully mineralised |
| T4 | Fully mineralised | Fully mineralised | Present, fully mineralised | Fully mineralised |
| T3 | Fully mineralised | Fully mineralised | Present, fully mineralised | Fully mineralised |
| T2 | Fully mineralised | Fully mineralised | Present, fully mineralised | Fully mineralised |
| T1 | Fully mineralised | Fully mineralised | Present, fully mineralised | Fully mineralised |
| *Phoebodus saidselachus* | | | | |
| T8 | Present, thin | Present, thin | Absent | Absent |
| T7 | Fully mineralised | Partially mineralised | Start of mineralisation | Start of mineralisation |
| T6 | Fully mineralised | Partially mineralised | Start of mineralisation | Start of mineralisation |
| T5 | Fully mineralised | Partially mineralised | Partially mineralised | Partially mineralised |
| T4 | Fully mineralised | Partially mineralised | Partially mineralised | Partially mineralised |
| T3 | Fully mineralised | Fully mineralised | Fully mineralised | Fully mineralised |
| T2 | Fully mineralised | Fully mineralised | Fully mineralised | Fully mineralised |
| T1 | Fully mineralised | Fully mineralised | Fully mineralised | Fully mineralised |

into the base both lingually and labially. As noted earlier, the mineralisation sequence in the teeth of *M. mohamezanei* is not as evident as in the aforementioned taxa. However, the presence of poorly mineralised trabecular dentine in the base of the youngest teeth suggests that its mineralisation pattern may more closely resemble that of *Ct. concinnus*, where trabecular dentine mineralises first in the crown and last in the base.

Thus, these major differences highlight a high disparity in dental mineralisation patterns, as previously described for neoselachians, occurring early in chondrichthyan evolution. Moreover, the observed similarities between chondrichthyan taxa that are phylogenetically and chronostratigraphically distant challenge the notion of a phylogenetic pattern.

Tooth replacement not only ensures the maintenance of a functional dentition but also facilitates the adaptation of tooth size and spacing to the overall body size of the animal during growth[7,8,10,41]. The tooth replacement rates for extinct chondrichthyans can be roughly estimated by examining tooth size differences within a single generative tooth file, in combination with comparative data from extant chondrichthyans. In the tooth files of *Ct. concinnus* and *M. saidselachus* there are notable size differences between younger and older teeth within the same file (Figs. 1, 2; Files 6 and 10[31]). Conversely, the teeth of *P. saidselachus* exhibit much smaller size differences. The following discussion provides rough estimates for tooth replacement rates based on these observations.

We adopt the methodology of Botella et al.[6] (Supplementary Note 3), who provided a mathematical framework for the estimation of tooth replacement in fossil taxa (in days/row) based on data from modern sharks. The tooth size increment ($\Delta s$) is estimated by measuring the size increase of teeth within a single tooth file (list with $\Delta s$ for some recent and extinct chondrichthyans given in Botella et al.[6]).

The resolution of CT data of *Ct. concinnus* and *P. saidselachus* does not allow for exact measurements. However, the thin sections of *Ct. concinnus* show a significant increase in tooth size, similar to other Devonian chondrichthyans[6]. Base width measurements derived from CT data of *M. mohamezanei* (Supplementary Fig. 3) yield a mean size increment of 12.04% between consecutive teeth. This value, although lower than other estimates for early chondrichthyans, remains notably higher than the size increments reported for extant neoselachians (mean 6%[6]). While the quality of the CT scan limits measurement precision, this estimate provides a general trend. Despite the limitations of this approach (see also Botella et al.[6]), the considerable size increase observed in consecutive teeth of *Ct. concinnus* and *M. saidselachus* suggest relatively slow tooth replacement rates, particularly when compared to *P. saidselachus* and even more so in modern sharks, thus corroborating previous observations[32]. The presence of strong tooth wear in *Ct. concinnus*[32], as well as in further early chondrichthyan taxa[6], further supports the presence of slow tooth replacement. In contrast, neoselachian teeth exhibit less wear[42] which we interpret as a consequence of their faster tooth replacement, leading to less food-to-tooth contact. The smaller tooth size differences observed in *P. saidselachus* suggest a higher tooth replacement rate, more akin to that of modern sharks. Therefore, while slow tooth replacement in comparison to modern sharks is not the general rule, it appears to be a common feature in many early chondrichthyan taxa.

Chimaeroids have evolved a specialised dentition composed of growing tooth plates[43,44] and thus lack the chondrichthyan characteristic of tooth replacement. These tooth plates are considered a derived character, having evolved through the fusion of tooth files from large but separate teeth in basal holocephalans[43]. These tooth plates are composed of hard, partially hypermineralised dentine, lacking enameloid[45,46]. Johanson et al.[47] state that tooth fusion represents a case of convergent evolution, occurring in

https://doi.org/10.1038/s42003-025-09320-0                                                                **Article**

Palaeozoic stem holocephalans, which exhibit varying degrees of tooth fusion or dental plate development.

We hypothesise that extended tooth retention constitutes an early precursor to the evolution of dental plates in the holocephalan lineage and constitutes the basis for tooth fusion. We base this on our findings in the CT scan from *M. mohamezanei* (File 9 and 10[31]), which revealed that in many cases, small remnants of teeth are located on the outside of the jaw (Figs. 1C, 2C). This is also seen in further stem holocephalans such as *Myriacanthus paradoxus*, as discussed in Johanson et al.[48] or *Cladoselache*, where these teeth are interpreted as old functional teeth, which are retained rather than being shed[8]. In *Maghriboselache*, the teeth are wrapped around the biting margin and retained on the outer, labial face of the jaw (Fig. 1C, 2C, Files 9 and 10[31]). However, a similar tooth retention pattern is observed in the stem elasmobranch genus *Ctenacanthus*, suggesting that this may be either a shared character among stem holocephalans or a plesiomorphic condition in chondrichthyans. The tooth arrangement of *M. mohamezanei* closely resembles that of *Helodus simplex*[44,47,49], which is regarded as the classical linking form between chimaeroids and their chondrichthyan ancestors[49]. *Helodus* is also the first genus to show evidence for tooth fusion within the holocephalan lineage. The largest to youngest teeth of the central tooth sets are fused into *"what might be considered rudimentary dental plates"*[49], while adjacent tooth sets remain unfused. Within the genus *Helodus*, there is a spectrum of fusion, ranging from closely arranged but unfused teeth to teeth fused only at the base, and finally, teeth that are fully fused[47]. We propose that the fusion of individual teeth of one tooth file, as in *Helodus simplex*[49], evolved from tooth retention. Labially retained teeth would have created spatial constraints for newly developing teeth, leading to more tightly spaced teeth and eventually to tooth fusion. Johanson et al.[47] state that fusion or incomplete separation of teeth within one file might be a trait retained from stem chondrichthyans, such as *Ptomacanthus* and *Doliodus*[50], and that there is no evidence for dental fusion in non-holocephalan crown chondrichthyans. Tooth retention is present in stem elasmobranchs (*Ctenacanthus*) as well as stem holocephalans (*Cladoselache, Maghriboselache*). This suggests that after the divergence of these lineages in the Silurian[51], tooth retention was lost in the elasmobranch lineage and a more frequent tooth replacement was established. In contrast, in the holocephalan lineage, tooth retention may have facilitated fusion, ultimately resulting in the evolution of the chimaeroid dentition, characterised by tooth plates.

## Methods

All three specimens presented herein were collected from the middle and late Famennian of the Moroccan Anti-Atlas by Mohammed Mezane (amateur palaeontologist from Merzouga, Morocco) and Saïd Oukherbouch (amateur palaeontologist from Tafraoute Sidi Ali, Morocco). Working permits were provided by the Ministère de l'Energie, des Mines, de l'Eau et de l'Environnement (Direction du Développement Minier, Division du Patrimoine, Rabat, Morocco). All fossils were exported to Switzerland and prepared at the Department of Paleontology of the University of Zurich. They are returned to Morocco and made available in the collections of the Higher School of Education and Training Berrechid (ESEFB), Hassan First University, Berrechid, Morocco.

Within the Anti-Atlas, sediments from the Precambrian to the Carboniferous are exposed along a large fold belt oriented in NE-SW direction[52]. The specimens were found in the Tafilalt and Maïder regions, where Devonian deposits are particularly well represented and crop out in E-W oriented synclines. These regions are renowned for rich and varied fossil assemblages, and several sites within the Tafilalt and Maïder basins qualify as Konzentrat- and Konservat-Lagerstätten, yielding diverse and exceptionally preserved fossil assemblages, including both invertebrates and vertebrates[29,30,32,53–58]. The vertebrate fauna comprises placoderms[55,56,58–62], osteichthyans[60,61,63,64], and chondrichthyans[29,30,53]. The Tafilalt and Maïder constitute two shallow marine basins during the Devonian, separated by the Tafilalt Platform, a structural high, that was locally emergent during the Late Devonian[52].

ESEFB-LTM-201, *Ctenacanthus concinnus*, was found in the Tafilalt at 31°00'57.4"N 4°03'58.2"W with no further chondrichthyan remains directly associated. It comprises two tooth files aligned on a fragment of the jaw cartilage. For a morphological description of the teeth, see Ginter et al.[23] or Greif et al.[32]. The position of the tooth files along the jaw is unknown (i.e., anterior or posterior). Whether the cartilage is part of the lower or the upper jaw cannot be determined. The teeth of both files are weathered at the surface.

ESEFB-LTM-203, *Phoebodus saidselachus* was found at Tizi Mouzgar (30.76937°N, 4.70647°W) near Madene El Mrakib in the Maïder. For a detailed description, see Frey et al.[29]. It preserves a complete three-dimensional skull embedded in sediment and encrusted by haematitic crusts. For a description of the species, see Frey et al.[29]. For the preparation of thin sections, a smaller piece was cut from the main block using a circular saw. This small block was taken between the upper and lower jaw and includes teeth of several files.

ESEFB-LTM-202, *Maghriboselache mohamezanei* was found in the Tafilalt at 30°57'40.2" 4°02'36.1"W. For a detailed description of the species, see Klug et al.[30]. The specimen preserves the entire three-dimensional skull. A smaller piece preserving two intact tooth files from the upper and lower jaw was cut and prepared for histological observations.

### Histological preparation

Thin sections with a thickness of about 60–100 µm were prepared following the procedure described in Chinsamy and Raath[65]. The sections were photographed using a Keyence digital-microscope VHX-7000 with a VHX-7100 fully integrated head, and a two-lens system (VHX-E20, VHX-E100). The images were adjusted in colour and contrast using Adobe Photoshop CS6 and Affinity Designer/Photo.

### X-ray computed tomography

All three specimens were scanned using the X-ray Industrial CT-Scanner at Qualitech in Mägenwil, Switzerland, with differing parameters: ESEFB-LTM-201, *Ctenacanthus concinnus*: 210 kV, 0.35 mA; voxel sizes in mm: x = y = z = 0.048 mm; total slices: 2211; 16-bit images (microfocus, Fxe). ESEFB-LTM-203, *Phoebodus saidselachus*: 215 kV, 0.35 mA; voxel sizes in mm: x =y =z = 0.056 mm; total slices: 1733; 16-bit images (microfocus, Fxe). ESEFB-LTM-202, *Maghriboselache mohamezanei*: 590 kV, 0.95 mA; voxel sizes in mm: x = y = z = 0.1068 mm; total slices: 3129; 16-bit images. Segmentation was performed in Mimics 26.0 (Files 5–10[31]).

### Terminology

For naming the tooth positions within the sections, we employ a tooth code adapted from previous works[5,9,18,19]. We use it in a simplified way, as we do not account for the position of the tooth file within the jaw (which is of subordinate importance due to the low degree of morphological variation of teeth within the jaw). Additionally, we do not label functional and replacement teeth separately, as we lack sufficient information to make this distinction. Instead, tooth number one is always assigned to the labial-most tooth. For *Maghriboselache mohamezanei*, the first two letters indicate whether the tooth file is from the upper jaw (Palatoquadrate, PQ) or lower jaw (Meckel's cartilage, MC). The last two digits in the tooth code indicate the position of the tooth within the file, starting with the labial-most tooth assigned the lowest number. MCT1, for example, refers to the labial-most tooth of the Meckel's cartilage. In the case of *Phoebodus saidselachus*, the distinction between the upper and lower jaws is unknown. Therefore, the first two letters describe the file, denoted from right to left as F1–F4, while the last two digits describe the position of the tooth within the file, T1–T8. For *Ctenacanthus concinnus*, only one tooth file is present in the sections. For this specimen, we simply describe the position of the tooth within the file from T1–T7, starting labially.

### Statistics and reproducibility

This study does not include any statistical analyses. Reproducibility is ensured as all materials used in this study are available with this publication and cited when necessary. The study is based on three specimens and 1–3

thin sections of each, which are stored in the collections of the Higher School of Education and Training Berrechid (ESEFB), Hassan First University, Berrechid, Morocco. Detailed photographs that are not figured here are provided online[31] and complement the figures and histological interpretations herein, ensuring the reproducibility of the study.

### Ethics and inclusion statement

This work is based on fossil material from the Anti-Atlas in Morocco and stems from long-lasting collaborations with local professional and amateur palaeontologists and researchers. Fossils are returned to Morocco and made available in the collections of the Higher School of Education and Training Berrechid (ESEFB) under the curation of Prof. Abdelouahed Lagnaoui, who is a close collaborator. Local collaboration enables the study of paleontologically important fossil material and helps highlight the palaeontological importance of the Moroccan Anti-Atlas. Working permits are provided by the Ministère de l'Energie, des Mines, de l'Eau et de l'Environnement (Direction du Développement Minier, Division du Patrimoine, Rabat, Morocco).

### Reporting summary

Further information on research design is available in the Nature Portfolio Reporting Summary linked to this article.

### Data availability

Thin sections of the specimens ESEFB-LTM-201, *Ctenacanthus concinnus*, ESEFB-LTM-202, *Maghriboselache mohamezanei* and ESEFB-LTM-203, *Phoebodus saidselachus*, are stored in the collections of the Higher School of Education and Training, Berrechid, Hassan First University, Berrechid, Morocco. Supplementary Information includes Supplementary Notes 1–3 and accompanying figures. Supplementary Information 1: Note on oral denticles and Figure depicting these in the thin sections of all three specimens. Supplementary Fig. 1: Histology of the oral denticles of all three specimens Supplementary Information 2 Note on the mineralisation sequence of the teeth of *Squatina*. Supplementary Fig. 2: Tooth file of *Squatina*. Supplementary Information 3: Note on size measurements and tooth replacement rates, including the equation used to estimate replacement for *Maghriboselache*. Supplementary Fig. 3D reconstruction of a tooth file of *Maghriboselache* showing base width measurements. Files 1–10 include unprocessed images and the CT data as well as 3D reconstructions of the specimens, stored on Zenodo: https://doi.org/10.5281/zenodo.15387379.

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

## Acknowledgements

We greatly acknowledge the colleagues from the Ministère de l'Energie, des Mines, de l'Eau et de l'Environnement (Direction du Développement Minier, Division du Patrimoine, Rabat, Morocco) for working and sample export

permits and the Swiss National Science Foundation (SNSF) for funding our work (grant nr 205320_215642). We thank Saïd Oukherbouch (Tafraoute, Morocco) and Mohamed Mezane (Merzouga, Morocco) for their hard work in the field, providing the fossils and for their long-lasting collaboration, as well as Prof. Abdelouahed Lagnaoui (ESEFB Berrechid, Morocco) for his support and collaboration. We thank René Kindlimann (Aathal, Switzerland) for fruitful chats regarding histology as well as further shark tooth-related topics. We thank Pedro Aquino from Eurofins Qualitech for the CT-scans and further advice regarding handling of the data, etc. Stefanie Herter (University of Zurich) kindly helped with the histological preparation of thin sections. We thank Irina Valeria Etter and Lena Sophie Gersbach (both University Zurich) for helping with the segmentation of *Maghriboselache* and *Phoebodus* in Mimics as part of their internships at the Department of Paleontology. And lastly, thanks to Amin El Fassi El Fehri (University of Zurich) for repeatedly reading through parts of the manuscript to improve language, as well as Jack L. Norton (University of Zurich) for proofreading the manuscript to improve the written English.

## Author contributions

M.G.: conceptualisation, data collection, investigation, methodology (preparation, photography, CT data), resources, validation, figures, writing – original draft, writing – review and editing; H.B.: conceptualisation, validation, writing – review and editing; T.S.: methodology (preparation), validation, writing – review and editing. C.K.: funding acquisition, investigation, resources, supervision, validation, writing – review and editing.

## Competing interests

The authors declare no competing interests.
