## [Transparent Peer Review file · Communications Biology]

Diversity of tooth mineralization patterns at the base of crown chondrichthyans

Corresponding Author: Ms Merle Greif

Version 0:

Reviewer comments:

Reviewer #1

(Remarks to the Author)
Dear Authors

Please find enclosed the review of manuscript entitled "Diversity of tooth mineralization patterns at the base of crown chondrichthyans", along with an annotated PDF file, including more specific comments and remarks about the manuscript.

General comments:

I must say it is really a nice work and should be appreciated. I have several but important comments which can improve the manuscript. I would suggest using proper scientific language throughout the manuscript, it looks very odd in a scientific paper. Although, the abstract and introduction is very informative, there is room for further improvement and clarity. I can see a lot of repetitions and stress on redundant information which can be reduced to minimum, please stress on the importance of your results and the evolutionary implications of the research.

I would have liked to see more discussion on histology and comparative analysis, which is important aspect of the paper. I guess you have cited many published work, there are series of published papers on the tooth histology, and you can compare your results with earlier works. I would stress more on the figure 7 which makes it important in evolutionary sense and it will enhance the comparison. The discussion section is also bit of worry for readers to understand and I believe there is enough room for improvement. In addition, there should be a separate paragraph where you can compare earlier histological findings (at least within the same age bracket) with your histological results. It will help to understand what new is being presented. I believe if presented in a clearer manner, it can improve the quality of the MS.

I would also recommend adding citation of the figures in the text properly and I believe the resolution of figures is good but can be improved, may be because of downsizing. The bibliography in the text and reference list probably also needs a bit more formatting – I also suggest changing some of the references and check they are cited at proper place in the text. Check properly how citations are referred in the text. You can not cite unpublished work unless the journal allows.

Otherwise, several smaller changes that should be very easy to perform can be found in the annotated PDF file. When all these edits are done, this manuscript will likely translate into a fine and well-organized study that would be of great interest to readers, which is why I recommend further revisions before acceptance of your manuscript.

Best wishes
Dr. Mohd Shafi Bhat

Reviewer #2

(Remarks to the Author)

The present work is a study of the dental histology of three late Devonian (Palaeozoic) cartilaginous fishes from Morocco, represented by more or less complete skeletons, i.e., *Ctenacanthus concinnus*, *Phoebodus saidselachus*, *Maghriboselache mohamezanei*. The authors state that while histological characters are largely studied in modern sharks, the neoselachians,

only little is known about these traits in their “earliest ancestors”, especially the sequence of mineralization (lines 31-33). For this, exceptional in situ preservation of teeth in articulation forming complete files is mandatory, which is provided by the three taxa forming the scope of the paper. The goals of this study therefore are to (1) fill the knowledge gap about the histology and mineralization processes of the teeth of early chondrichthyans, (2) address tooth replacement rates and modes of these taxa in comparison to modern analogues and (3) discuss both characteristics, i.e., tooth histology and tooth replacement mode in the context of chondrichthyan phylogeny as these traits enabled chondrichthyans to inhabit a large variety of trophic niches and ultimately to survive four major mass extinction events (lines 68-74).

Comment 1. I am not a native speaker, but the writing style sounds very bumpy, and the grammar should be checked by a native speaker.

Comment 2. The references to the supplementary material is difficult to follow as you included three different suppl. materials, but you do not refer to any specific one throughout the text. This needs to be changed, because otherwise it is not possible to follow your arguments with a lot of effort trying to identify the correct suppl. material!

The authors continue to briefly describe the histology and tooth replacement mode in all three taxa separately. Unfortunately, they do not provide any higher systematic framework for the three taxa so that it remains ambiguous what these represent, especially as the authors call these three taxa to be “chondrichthyan key species” (line 65) and refer to them as “stem-group representatives” (lines 225, 254) and “extinct lineages at the stem of crown chondrichthyans” (line 254).

Comment 3. I would strongly suggest deleting ‘extinct lineages’ (line 254) as the stem always represents extinct taxa per definition and thus is a tautology.

Comment 4. The authors seemingly assume that all three taxa represent stem chondrichthyans, i.e., on the stem leading to the divergence of euchondrocephalans and elasmobranchs. However, *Ctenacanthus* and *Phoebodus* are considered to be stem elasmobranchs while *Maghriboselache* represents a stem euchondrocephalan, maybe even a stem holocephalan. So, we are dealing with two different evolutionary lineages. The authors, however, do not distinguish between major systematic units and do not provide any phylogenetic framework to put the results in a useful context as claimed in lines 68-74 (see above). Therefore, the authors fail to provide any useful discussion in relation to the third goal mentioned in the corresponding section (see also above, first paragraph of evaluation). I would strongly suggest putting all into a solid phylogenetic and systematic framework and to present the results and discussions combined, not separated for each taxon, i.e., describing the pattern in the stem euchondrocephalan / holocephalan and discuss it in a broad systematic context and describing the pattern in the stem elasmobranchs and discussing it in broad systematic context, respectively. The division into each taxon makes it probably useful from a strict descriptive point of view but is not useful for broader, global perspectives. This means, the structure of the manuscript needs to be drastic changed to make it a study not only interesting for some specialists but for a broader scientific community.

The authors argue that a better understanding of tooth histologies in Palaeozoic sharks and knowledge about the mineralization sequence might “reveal important information regarding the evolution of a dentition that enables modern sharks to cover a wide range of trophic levels” (line 33-34). However, as the patterns are not discussed in a broad phylogenetic / systematic context, the authors fail to provide valid information addressing this goal.

Comment 5. The authors state that in neoselachians fast tooth replacement is combined with highly differentiated tooth histology including a triple layered enameloid cap indicating a close implying that both are closely linked (lines 50-51). The references the authors cite for this dependence, however, do not indicate such a relationship to my knowledge! A similar claim is made for plesiomorphic sharks with simpler histology, which again is not supported by the references. All the references are referring to the histology but not to speed of tooth replacement being dependant on tooth histology. Rather, speed of tooth replacement is more likely related to the amount of calcium, phosphate and other elements to form teeth or other skeletal parts, respectively. Please check this claim and re-phrase this part if necessary.

Comment 6. The authors state that there is a difference of tooth replacement pattern between ancient and modern sharks, referring to “fast” versus “slow” (lines 50 ff.). What does “fast” and “slow” mean? These terms are very relative. You should provide examples for what is “fast” and “slow” (speed of tooth replacement differs strongly within living sharks).

Comment 7. You indicate that tooth size increases from labial to lingual within a file (line 85), which relates to ontogeny. However, this needs to be quantified in your study as you intend to use it as a measure for tooth replacement rate in your study. There are now several papers available with corresponding CT scans that could be used to measure tooth size along files in extant taxa, which subsequently could be used for comparison with your Palaeozoic taxa to verify your interpretation.

Comment 8. You note that there is no indication of enameloid differentiation identifiable in your scans and sections at any magnification (lines 88-89, lines 143-144, line 223). However, the magnification of the CT scans as well as the figure size of the sectioned specimens are not good enough to verify this. Please provide a high-resolution magnification of one of the scans, probably using polarized light to demonstrate that there really is no differentiation. However, the best way to identify the enameloid pattern is using etched surfaces of sectioned teeth for evaluation with a scanning electron microscope. Therefore, your conclusion is not 100% conclusive, even though we can expect it to be a single crystallite enameloid.

Comment 9. In lines 115-118 the authors state that “Deposition of osteodentine starts very early as well. It grows “rapidly” from the apex of the cusps towards the base (in tooth T7, Fig.2 osteodentine is already quite mineralized at the apex of the

cusps but tapers out towards the base). The rapid occupation of the cusp by osteodentine stops the centripetal growth of orthodentine". This statement is quite speculative as it is unclear what "rapidly" is. How do you know that it is rapid, especially the "occupation" [maybe better "filling"] of the cusp? It also could be slow assuming that tooth replacement in these plesiomorphic chondrichthyans is slow.

Comment 10. You note that osteodentine formation starts suddenly, while enameloid and orthodentine might start to form simultaneously. To my knowledge, this is not exactly correct as enameloid always starts first to develop, as enamel in mammals, and orthodentine / osteodentine starts successively (lines 56-60). Please provide exact examples for taxa in which it has been demonstrated that both, enameloid and orthodentine start forming at the same time.

Comment 11: Teeth in chondrichthyan fishes are never anchored / fused to the jaw. So, how can they become separated from the jaw cartilage during mineralization? (lines 120-121)

Comment 12. Dentine is very diverse tissue with many different types, especially in early vertebrates. Generally, three types can be distinguished:

1. Circumpulpar dentine: surrounding a pulp cavity and is analogous to circumvascular dentine located around a vascular canal
2. Pallial dentine: is a thin layer immediately exterior to circumpulpar or circumvascular dentine, which can be distinguished from enameloid by the presence of dentinal tubes
3. Tubular dentine: consists of dentinal osteons separated by an enamel(oid)-like interosteonal material without any outer layer of pallial dentine or enamel(oid)

You are seemingly using "orthodentine" and "pallial dentine" interchangeably (e.g., line 146), as you do with "osteodentine" and "trabecular dentine" (line 30). Simply that orthodentine occurs in what you call "pallial part of the cusp" (line 241) does not make it automatically to "pallial dentine". Please use the terms appropriately according to their definitions!

Comment 13. In lines 242-247, you state that the tooth histology of *Phoebodus saisedelachus* differs from all other in that the orthodentine is extending far more basally compared to the others (see also Fig. 3C). The teeth of *Phoebodus* are tricuspid. Which cusp did you section? And could it simply be that you have an oblique section where you see the orthodentine extending towards the adjacent cusp? The strange form of the cusp would suggest such an oblique section. It would be necessary to have a sagittal section ideally through the middle cusp to really identify the basal extension of the orthodentine. You should compare your results to previously published ones, such as, e.g., Ivanov (1999), where no distinct histological type is evident. Also, the statement that the pulp cavity is only small and restricted to the apical part (line 243) as well as the enameloid being restricted to the upper part of the crown (line 250) could be related to obliqueness of the section, and thus an artefact. Since your interpretation is important, it is necessary to take very much care of the sectional plane!

Comment 14. By now, three histotypes of chondrichthyan teeth are distinguished, depending on the presence or absence of orthodentine and osteodentine in the crown, which always is covered by a superficial, highly mineralized enameloid (compare Jambura et al. 2018, 2019, 2020):

1. Orthodont teeth: central hollow pulp cavity present encapsulated by a substantial layer of orthodentine
2. Pseudoosteodont teeth: osteodentine intrudes from the root into the hollow pulp cavity, which in fully mineralized teeth is replaced by an osteodont core that is surrounded by orthodentine
3. Osteodont teeth: no orthodentine is developed, but the complete dentinal core of the crown consists of osteodentine and replaces the hollow pulp cavity

But the presence or absence of an open pulp cavity (lines 234-247) is no distinguishing character at all!

Comment 15. Chapter "Sequence of mineralization" (lines 282). This section needs to be completely re-arranged (see comment Comment 4).

Comment 16. In Lines 287-288 and 327-331, you indicate that the major difference in mineralization pattern is related to *Phoebodus*, as the orthodentine extends far basally (but see above comment about section direction) and those listed in Table 1. The major difference in Table 1 is that osteodentine is absent in the crown in T8 in *Phoebodus*, while it is present and densifying in T8 in *Maghriboselache*. This replacement tooth (T8) is not present in *Ctenacanthus*. I wonder if this really justifies assuming that mineralization in *Phoebodus* is distinct and unique?

Comment 17. What is the difference between "densifying" and "porous" as listed in Table 1

Comment 18. You state that there are visible size differences in teeth of the same family between *Ctenacanthus* and *Maghriboselache*. I honestly do not see these size differences. Such a statement needs to be supported by quantified data!

Comment 19. Lines 351-356. Please insert a table with the individual measurement of the teeth so that the values you are giving for size increase along the tooth files can be reproduced.

Comment 20. In Figure 7, you indicate that in xenacanthids, the crown can consist of pleromin. But pleromin is a very specialized tissue, which only is present in the tooth plates of holocephalans! Please provide references for your histology interpretations in the cladogram.

Comment 21. Figure 7. Please indicate to which clade the three taxa described in this study belong.

Comment 22. According to the letter of the Directorate of Geology of the corresponding Moroccan ministry you had permission to collect rock and fossil samples. But it also states: "The export of collected rock samples and fossils requires verification and approval by the Directorate of Geology." I wonder if you have a permission that explicitly allows you to export the fossil specimens described in this paper to Zürich and to permanently house these in the Zürich museum? I would have expected that all collected material has to be returned to a central, leading museum in Morocco, such as the Marrakech Natural History Museum.

I have include some more comments, suggestions and corrections, respectively, directly in the text (see attachment), which I would like to aks you to consider.

I hope these comments are helpful. Please feel free to contact me if you have any questions.

Kind regards,

Jürgen Kriwet

Reviewer #3

(Remarks to the Author)

I think that this is a really nicely constructed and really useful paper on the origins of chondrichthyan dentitions. The specimens used are very good for this study and the imaging is of fantastic quality. The descriptions of the material and the structures of the teeth are very clear and show a continuity with the structure of teeth of more modern taxa. I think there could be a little more on the implications of this, and if there is a need for space to do so, maybe some of the descriptions of microstructures where they are similar in different taxa could be condensed. There could maybe be more made of the unique position of the pulp cavity in phoebodonts. Are these the most basal chondrichthyes to have one? If so what are the implications for evolution of this- did it start near the tip of the tooth and move down to the top of the osteodentine later? The retention of the early teeth on post functional positions is, as said, clearly holocephalan-like. This was discussed a bit in Johanson et al 2021 "The stem-holocephalan Helodus (Chondrichthyes; Holocephali) and the evolution of modern chimaeroid dentitions" where it is also noted that some other basal members of the holocephalan clade have teeth that are basally fused; this does not seem to be the case in the species figured here but does seem to be present in Doliodus. It would be useful if there was a discussion of the timing, and potentially mechanisms, for change between fused and unfused teeth, and whither there was a reversal of state and/or multiple events of teeth fusion.

I assume you are using the term "elasmobranch" for the entire "shark clade" (ie closer to say Carcharias and Raja than Chimaera), as opposed to Maisey's restricted use. Maybe you should make this clear.

Version 1:

Reviewer comments:

Reviewer #1

(Remarks to the Author)

Dear Authors

Please find enclosed the review of manuscript entitled "Diversity of tooth mineralization patterns at the base of crown chondrichthyans", along with an annotated PDF file, including remarks about the manuscript.

I congratulate you for your study and thank you for considering to revise the manuscript as advised. I enjoyed the manuscript and it has been improved immensely from the earlier version. Minor correction that should be very easy to perform can be found in the annotated PDF file. When all these edits are done, this manuscript will likely translate into a fine and well-organized study that would be of great interest to readers, I have recommended the acceptance of your manuscript with few typos or minor edits.

Best wishes

Dr. Mohd Shafi Bhat

Reviewer #2

(Remarks to the Author)

Dear authors,

Thank you very much for considering all comments and suggestions provided by me, also those that you don't exactly agree on. I am nevertheless glad to see the changes and improvements you made on the manuscript and I don't have any further

suggestions. I am sure that this study contributes importantly to our understanding of the histology in stem members of different stem chondrichthyans.

Kind regards,
Jürgen Kriwet

Rebuttal Letter for

Manuscript COMMSBIO-25-5832-T

Diversity of tooth mineralisation patterns at the base of crown chondrichthyans

Dear Editor and Reviewers,

Many thanks for the thorough reviews. We have revised our manuscript following the comments and suggestions given by all three reviewers. One of the main concerns voiced by Reviewer 2 was that the phylogenetic and systematic relationships were not clearly stated and discussed. We have focused on providing this information already in the Introduction as well as discussing it further. This led to major changes in large parts of the manuscript and highlights the importance of Figure 7. Two reviewers mentioned that the written English sounds “bumpy”. It was revised by a native speaker from the UK.

All corrections and suggestions given by reviewers 1 and 2 in the annotated files were considered and our responses are given directly as comments in the marked-up manuscript file. According to the Reviewers’ comments large parts of the manuscript have been rewritten.

Following the comments of reviewer 1, we have merged the results parts of the three separate chapters for each species each called “Sequence of mineralisation during tooth development” following their histological description into one. This chapter describes and compares mineralisation of all three taxa. According to Reviewer 1 *“results were mixed up” and he commented: “This is again confusing, it does not make any sense (I would write a single paragraph for all three species and describe and compare mineralisation stages)”*. We followed this advice. However, we are not sure if this is an improvement or not. We think the results are more mixed up now, but we are open to keep the new version if deemed better.

Reviewer 2 additionally pointed out (comment 22) that the given permit only allowed for collection of rock and fossil samples while the export of collected rock samples and fossils requires verification and approval by the Directorate of Geology. This is correct. Due to the sudden lockdown caused by Covid-19 we had to leave Morocco in a rush in that year and there was no time to get back to the Directorate of Geology to get this permission (as usually done in other years). We have been in touch with our colleague in Berrechid, Morocco and decided to store two of the specimens in the collections of the Higher School of Education and Training Berrechid (ESEFB) of the Hassan First University, Berrechid, Morocco. The third specimen, *Phoebodus* is stored in Cadi-Ayyad University, Marrakesh, Morocco.

We have encountered some contradictions between the reviews. While reviewer 1 asks to give only the “reference” for supplementary material and no additional information, reviewer 2 points out that it was hard to follow the supplementary material without any indication of which exact file. We decided to number all online material on Zenodo and give those numbers together with the DOI. Supplementary material that is not on Zenodo is referred to as supplementary information.

Furthermore, reviewer 3 pointed out that the quality of the Figures is excellent while reviewer 1 stated that the resolution can be improved. We have excellent, high-resolution original versions of all Figures. They will be submitted herewith.

Please find below our detailed answers to the reviewer comments. Reviewer comments are given in black, our answers in blue:

Reviewer #1 (Remarks to the Author):

Dear Authors

Please find enclosed the review of manuscript entitled "Diversity of tooth mineralization patterns at the base of crown chondrichthyans", along with an annotated PDF file, including more specific comments and remarks about the manuscript.

General comments:

I must say it is really a nice work and should be appreciated. I have several but important comments which can improve the manuscript. I would suggest using proper scientific language throughout the manuscript, it looks very odd in a scientific paper. Although, the abstract and introduction is very informative, there is room for further improvement and clarity. I can see a lot of repetitions and stress on redundant information which can be reduced to minimum, please stress on the importance of your results and the evolutionary implications of the research.

***Thank you very much!

We have reorganized parts of the Introduction in order to minimize repetitions and to add some more important information that had not been mentioned yet.

I would have liked to see more discussion on histology and comparative analysis, which is important aspect of the paper. I guess you have cited many published work, there are series of published papers on the tooth histology, and you can compare your results with earlier works. I would stress more on the figure 7 which makes it important in evolutionary sense and it will enhance the comparison. The discussion section is also bit of worry for readers to understand and I believe there is enough room for improvement. In addition, there should be a separate paragraph where you can compare earlier histological findings (at least within the same age bracket) with your histological results. It will help to understand what new is being presented. I believe if presented in a clearer manner, it can improve the quality of the MS.

***We compare histological findings in the Discussion part under the heading "Histology of stem chondrichthyans". We furthermore focused on providing more information on the phylogenetic relationships among the studied taxa and by doing so, we highlighted the importance of Figure 7.

I would also recommend adding citation of the figures in the text properly and I believe the resolution of figures is good but can be improved, may be because of downsizing. The bibliography in the text and reference list probably also needs a bit more formatting – I also suggest changing some of the references and check they are cited at proper place in the text. Check properly how citations are referred in the text. You can not cite unpublished work unless the journal allows.

*** As mentioned above, we have excellent, high-resolution original versions of all Figures. They will be submitted herewith. We have removed citations for unpublished work.

Otherwise, several smaller changes that should be very easy to perform can be found in the annotated PDF file. When all these edits are done, this manuscript will likely translate into a fine and well-organized study that would be of great interest to readers, which is why I recommend further revisions before acceptance of your manuscript.

***We have addressed the comments given in the annotated PDF. Accordingly, we have

restructured large parts of the manuscript and streamlined the discussion. We have merged the results parts, sequence of mineralisation, into one part for all three taxa and we have changed parts of the introduction.

Best wishes
Dr. Mohd Shafi Bhat

Reviewer #2 (Remarks to the Author):

The present work is a study of the dental histology of three late Devonian (Palaeozoic) cartilaginous fishes from Morocco, represented by more or less complete skeletons, i.e., *Ctenacanthus concinnus*, *Phoebodus saidselachus*, *Maghriboselache mohamezanei*. The authors state that while histological characters are largely studied in modern sharks, the neoselachians, only little is known about these traits in their “earliest ancestors”, especially the sequence of mineralization (lines 31-33). For this, exceptional in situ preservation of teeth in articulation forming complete files is mandatory, which is provided by the three taxa forming the scope of the paper. The goals of this study therefore are to (1) fill the knowledge gap about the histology and mineralization processes of the teeth of early chondrichthyans, (2) address tooth replacement rates and modes of these taxa in comparison to modern analogues and (3) discuss both characteristics, i.e., tooth histology and tooth replacement mode in the context of chondrichthyan phylogeny as these traits enabled chondrichthyans to inhabit a large variety of trophic niches and ultimately to survive four major mass extinction events (lines 68-74).

Comment 1. I am not a native speaker, but the writing style sounds very bumpy, and the grammar should be checked by a native speaker.

***The writing and grammar has been revised by a native speaker (Jack Norton from the UK, a colleague working at the University of Zurich).

Comment 2. The references to the supplementary material is difficult to follow as you included three different suppl. materials, but you do not refer to any specific one throughout the text. This needs to be changed, because otherwise it is not possible to follow your arguments with a lot of effort trying to identify the correct suppl. material!

***We have included more information about which supplementary material is cited each time we refer to a supplement. Each supplementary file on Zenodo is numbered now and these numbers are given in the main text. Whenever we felt it was necessary, we provided more information about the supplement (e.g. Photo ID or short ID) to facilitate finding the corresponding supplementary data.

We are not sure why the Reviewer mentions three different supplementary materials. There is one file that includes supplementary information in a word file. Each information given therein is cited with the according headers in the text. And there is supplementary material at Zenodo, which comprises ten files (now numbered). Whenever referred to this material, the DOI is given.

The authors continue to briefly describe the histology and tooth replacement mode in all three taxa separately. Unfortunately, they do not provide any higher systematic framework

for the three taxa so that it remains ambiguous what these represent, especially as the authors call these three taxa to be “chondrichthyan key species” (line 65) and refer to them as “stem-group representatives” (lines 225, 254) and “extinct lineages at the stem of crown chondrichthyans” (line 254).

***We have focused more on describing the phylogenetic context and discussing the findings accordingly. In this regard, we now introduce the phylogenetic position of the three discussed taxa already in the introduction. Throughout the manuscript, we discuss our histological findings in a clearer phylogenetic context. We also highlight Figure 7, which reflects these relationships.

Comment 3. I would strongly suggest deleting ‘extinct lineages’ (line 254) as the stem always represents extinct taxa per definition and thus is a tautology.

***deleted

Comment 4. The authors seemingly assume that all three taxa represent stem chondrichthyans, i.e., on the stem leading to the divergence of euchondrocephalans and elasmobranchs. However, *Ctenacanthus* and *Phoebodus* are considered to be stem elasmobranchs while *Maghriboselache* represents a stem euchondrocephalan, maybe even a stem holocephalan. So, we are dealing with two different evolutionary lineages. The authors, however, do not distinguish between major systematic units and do not provide any phylogenetic framework to put the results in a useful context as claimed in lines 68-74 (see above). Therefore, the authors fail to provide any useful discussion in relation to the third goal mentioned in the corresponding section (see also above, first paragraph of evaluation). I would strongly suggest putting all into a solid phylogenetic and systematic framework and to present the results and discussions combined, not separated for each taxon, i.e., describing the pattern in the stem euchondrocephalan / holocephalan and discuss it in a broad systematic context and describing the pattern in the stem elasmobranchs and discussing it in broad systematic context, respectively. The division into each taxon makes it probably useful from a strict descriptive point of view but is not useful for broader, global perspectives. This means, the structure of the manuscript needs to be drastic changed to make it a study not only interesting for some specialists but for a broader scientific community.

***We have changed large parts of the manuscript in this regard (see above). However, even in the first manuscript version, it was clearly stated (several times) that *Maghriboselache* is a stem-holocephalan while the other taxa are stem elasmobranchs. Discussing the findings for stem holocephalans and stem elasmobranchs separately does not make a lot of sense since there is no phylogenetic link reflected in the histological patterns. This is clearly stated as well.

The authors argue that a better understanding of tooth histologies in Palaeozoic sharks and knowledge about the mineralization sequence might “reveal important information regarding the evolution of a dentition that enables modern sharks to cover a wide range of trophic levels” (line 33-34). However, as the patterns are not discussed in a broad phylogenetic / systematic context, the authors fail to provide valid information addressing this goal.

***This part was largely rephrased. However, the overall aim remains. We have added some general information regarding phylogenetic positions of the three examined taxa in the introduction and discussed this in greater detail (see above).

Comment 5. The authors state that in neoselachians fast tooth replacement is combined with highly differentiated tooth histology including a triple layered enameloid cap indicating a close implying that both are closely linked (lines 50-51). The references the authors cite for this

dependence, however, do not indicate such a relationship to my knowledge! A similar claim is made for plesiomorphic sharks with simpler histology, which again is not supported by the references. All the references are referring to the histology but not to speed of tooth replacement being dependant on tooth histology. Rather, speed of tooth replacement is more likely related to the amount of calcium, phosphate and other elements to form teeth or other skeletal parts, respectively. Please check this claim and re-phrase this part if necessary.

***The statements made in these lines were admittedly misleading. We rephrased them. It was not intended to state that histology and tooth replacement depend on each other. We simply wanted to point out (1) differences in tooth replacement rates between neoselachians and early chondrichthyans and (2) differences in the histology of neoselachians and early chondrichthyans. In addition to these topics, we added a short explanation as to why tooth replacement rates are generally assumed to be slower in early chondrichthyans than in modern sharks. This is, however, repeated in more detail in the discussion part "Tooth replacement rates and tooth retention".

Comment 6. The authors state that there is a difference of tooth replacement pattern between ancient and modern sharks, referring to "fast" versus "slow" (lines 50 ff.). What does "fast" and "slow" mean? These terms are very relative. You should provide examples for what is "fast" and "slow" (speed of tooth replacement differs strongly within living sharks).

***As stated above, we added a short explanation, and this is furthermore discussed in the discussion part "Tooth replacement rates and tooth retention". We refrain from giving total numbers for tooth replacement rates for early chondrichthyans since these are relative and there are no methods yet to reconstruct absolute rates.

Comment 7. You indicate that tooth size increases from labial to lingual within a file (line 85), which relates to ontogeny. However, this needs to be quantified in your study as you intend to use it as a measure for tooth replacement rate in your study. There are now several papers available with corresponding CT scans that could be used to measure tooth size along files in extant taxa, which subsequently could be used for comparison with your Palaeozoic taxa to verify your interpretation.

***We are following the approach in Botella et al. (2009: Tooth replacement rates in early chondrichthyans: a qualitative approach. *Lethaia* **42**, 365–376) Since the focus of the manuscript is on the histology and mineralisation patterns of teeth in early chondrichthyans, we think that it is out of scope to add such kind of table. This would require collecting a big amount of data. We rephrased the according parts of the manuscript to clarify that we are only making rough estimates and discuss these. However, we stress that replacement rates in many early chondrichthyans are slower than in many modern elasmobranchs based on the size differences of teeth that can be measured in our specimens in combination with sometimes strong tooth wear.

Comment 8. You note that there is no indication of enameloid differentiation identifiable in your scans and sections at any magnification (lines 88-89, lines 143-144, line 223). However, the magnification of the CT scans as well as the figure size of the sectioned specimens are not good enough to verify this. Please provide a high-resolution magnification of one of the **scans**, probably using polarized light to demonstrate that there really is no differentiation. However, the best way to identify the enameloid pattern is using etched surfaces of sectioned teeth for evaluation with a scanning electron microscope. Therefore, your conclusion is not 100% conclusive, even though we can expect it to be a single crystallite enameloid.

***We state that, to be 100% conclusive, we would need more information discussion: *"However, to definitely exclude differentiation of enameloid as seen in neoselachians (i.e.,*

*layered enameloid*¹⁶), further examinations using a scanning electron microscope and acidic etching would be needed (as performed in Gillis and Donoghue (2007)¹⁷ or Singh et al.³⁴).

Comment 9. In lines 115-118 the authors state that “Deposition of osteodentine starts very early as well. It grows “rapidly” from the apex of the cusps towards the base (in tooth T7, Fig.2 osteodentine is already quite mineralized at the apex of the cusp but tapers out towards the base). The rapid occupation of the cusp by osteodentine stops the centripetal growth of orthodentine”. This statement is quite speculative as it is unclear what “rapidly” is. How do you know that it is rapid, especially the “occupation” [maybe better “filling”] of the cusp? It also could be slow assuming that tooth replacement in these plesiomorphic chondrichthyans is slow.

***This part was largely rephrased; the most important point here is that the orthodentine mineralised before the osteodentine and that the osteodentine limits the inward growth of orthodentine. Rapid is relative, we clarified this.

Comment 10. You note that osteodentine formation starts suddenly, while enameloid and orthodentine might start to form simultaneously. To my knowledge, this is not exactly correct as enameloid always starts first to develop, as enamel in mammals, and orthodentine / osteodentine starts successively (lines 56-60). Please provide exact examples for taxa in which it has been demonstrated that both, enameloid and orthodentine start forming at the same time.

***We agree. It is always the first to mineralise. We have clarified this throughout the manuscript.

Comment 11: Teeth in chondrichthyan fishes are never anchored / fused to the jaw. So, how can they become separated from the jaw cartilage during mineralization? (lines 120-121)

***This part was poorly phrased. We rewrote it and mention that at the onset of mineralisation in the cusp is located close to the jaw cartilage and that throughout mineralisation and due to the onset of mineralisation of the tooth base, the tooth moves further away from the jaw as visible in Fig 2A.

Comment 12. Dentine is very diverse tissue with many different types, especially in early vertebrates. Generally, three types can be distinguished:

1. Circumpulpar dentine: surrounding a pulp cavity and is analogous to circumvascular dentine located around a vascular canal
2. Pallial dentine: is a thin layer immediately exterior to circumpulpar or circumvascular dentine, which can be distinguished from enameloid by the presence of dentinal tubes
3. Tubular dentine: consists of dentinal osteons separated by an enamel(oid)-like interosteonal material without any outer layer of pallial dentine or enamel(oid)

You are seemingly using “orthodentine” and “pallial dentine” interchangeable (e.g., line 146), as you do with “osteodentine” and “trabecular dentine” (line 30). Simply that orthodentine occurs in what you call “pallial part of the cusp” (line 241) does not make it automatically to “pallial dentine”. Please use the terms appropriately according to their definitions!

*** We only distinguish two types of dentine, trabecular dentine and orthodentine. Pallial only refers to the deposition (it was adapted throughout the manuscript). The term osteodentine has been removed throughout the manuscript (except for direct citations) and replaced with trabecular dentine. We prefer this term since it is the original term that describes this type of material more precisely. Osteodentine, in contrast, is often used as a synonym for trabecular dentine but technically refers to a material that contains bone cells. According to Peyer

(1968) this is not the case for Chondrichthyans. This statement was added in the Discussion part “Histotype scheme and its limitations”.

Comment 13. In lines 242-247, you state that the tooth histology of *Phoebodus saisedelachus* differs from all other in that the orthodontine is extending far more basally compared to the others (see also Fig. 3C). The teeth of *Phoebodus* are tricuspid. Which cusp did you section? And could it simply be that you have an oblique section where you see the orthodontine extending towards the adjacent cusp? The strange form of the cusp would suggest such an oblique section. It would be necessary to have a sagittal section ideally through the middle cusp to really identify the basal extension of the orthodontine. You should compare your results to previously published ones, such as, e.g., Ivanov (1999), where no distinct histological type is evident. Also, the statement that the pulp cavity is only small and restricted to the apical part (line 243) as well as the enameloid being restricted to the upper part of the crown (line 250) could be related to obliqueness of the section, and thus an artefact. Since your interpretation is important, it is necessary to take very much care of the sectional plane!

***We are confident about the correctness of our description (Fig. 3C). In fact, it is similar to further findings in phoebodonts. We are discussing these findings and additionally, Figure 3 of the supplementary material shows the modern shark *Squatina* displaying a similar histology. Due to the nature of the fossil material that we had available, the section was taken in an angle, this is noted in the manuscript.

Comment 14. By now, three histotypes of chondrichthyan teeth are distinguished, depending on the presence or absence of orthodontine and osteodontine in the crown, which always is covered by a superficial, highly mineralized enameloid (compare Jambura et al. 2018, 2019, 2020):

1. Orthodont teeth: central hollow pulp cavity present encapsulated by a substantial layer of orthodontine
2. Pseudoosteodont teeth: osteodontine intrudes from the root into the hollow pulp cavity, which in fully mineralized teeth is replaced by an osteodont core that is surrounded by orthodontine
3. Osteodont teeth: no orthodontine is developed, but the complete dentinal core of the crown consists of osteodontine and replaces the **hollow** pulp cavity

But the presence or absence of an **open** pulp cavity (lines 234-247) is no distinguishing character at all!

***All three types are described in the discussion part “Histotype scheme and its limitations”. We are subsequently discussing the presence or absence of a **hollow** pulp cavity, not **open**.

However, we agree that the most important character is not the absence or presence of the cavity but the presence and arrangement of orthodontine and osteodontine as we had written in the discussion.

Comment 15. Chapter “Sequence of mineralization” (lines 282). This section needs to be complete re-arranged (see comment Comment 4).

***It has been largely rearranged.

Comment 16. In Lines 287-288 and 327-331, you indicate that the major difference in mineralization pattern is related to *Phoebodus*, as the orthodontine extends far basally (but see above comment about section direction) and those listed in Table 1. The major difference in Table 1 is that osteodontine is absent in the crown in T8 in *Phoebodus*, while it

is present and densifying in T8 in *Maghriboselache*. This replacement tooth (T8) is not present in *Ctenacanthus*. I wonder if this really justifies assuming that mineralization in *Phoebodus* is distinct and unique?

***As we have pointed out in earlier comments, the most important aspect is that orthodontine extends far down towards the base. This is not an artefact and can be seen in other phoebodonts. This is discussed in the manuscript.

Comment 17. What is the difference between “densifying” and “porous” as listed in Table 1

***We have changed the terms used in the table to according to a similar table given in Schnetzet al. (2016: Tooth development and histology patterns in lamniform sharks (Elasmobranchii, Lamniformes) revisited. *J. Morphol.* **277**, 1584–1598).

Comment 18. You state that there are visible size difference in teeth of the same family between *Ctenacanthus* and *Maghriboselache*. I honestly do not see these size differences. Such a statement needs to be supported by quantified data!

*** We think there might be a misunderstanding. We are not stating that these two files differ in size but the teeth of each file. We rephrased this for clarification.

Furthermore, the differences in tooth size within each file in each taxon is well discernible in Fig.1. It is also shown in the supplementary Information (Supplementary Information 2, Fig.2). The tooth base width ranged from 5.04 mm to 11.0 mm. We consider this a significant size difference.

Comment 19. Lines 351-356. Please insert a table with the individual measurement of the teeth so that the values you are giving for size increase along the tooth files can be reproduced.

***The exact measurements, including a picture of the tooth file in *Maghriboselache*, are provided in the supplementary material as cited in the text.

Comment 20. In Figure 7, you indicate that in xenacanthids, the crown can consist of pleromin. But pleromin is a very specialized tissue, which only is present in the tooth plates of holocephalans! Please provide references for your histology interpretations in the cladogram.

***This was a misinformation, we apologize for this error. Figure 7 was adapted, and the references are given in a table in the supplementary material (Supplementary information 4, Table 1). This is now mentioned in the caption.

Comment 21. Figure 7. Please indicate to which clade the three taxa described in this study belong.

*** Figure 7 was updated. To clarify where each taxon belongs, we added their names to the histological pictures that are given with their phylogenetic relation. The caption was adapted as well (see comment above).

Comment 22. According to the letter of the Directorate of Geology of the corresponding Moroccan ministry you had permission to collect rock and fossil samples. But it also states: "The export of collected rock samples and fossils requires verification and approval by the Directorate of Geology." I wonder if you have a permission that explicitly allows you to export the fossil specimens described in this paper to Zürich and to permanently house these in the Zürich museum? I would have expected that all collected material has to be returned to a

central, leading museum in Morocco, such as the Marrakech Natural History Museum.

*** As mentioned at the start, this is correct. Due to the sudden lockdown caused by Covid-19 we had to leave Morocco in a rush and there was no time to get back to the Directorate of Geology to get this permission (as usually done in other years). Two of the samples are now stored in the collections of Berrechid, the third one is stored in the collections of Marrakesh.

I have include some more comments, suggestions and corrections, respectively, directly in the text (see attachment), which I would like to aks you to consider.

***We changed the text and all comments were considered.

I hope these comments are helpful. Please feel free to contact me if you have any questions.

Kind regards,

Jürgen Kriwet

Reviewer #3 (Remarks to the Author):

I think that this is a really nicely constructed and really useful paper on the origins of chondrichthyan dentitions. The specimens used are very good for this study and the imaging is of fantastic quality. The descriptions of the material and the structures of the teeth are very clear and show a continuity with the structure of teeth of more modern taxa.

I think there could be a little more on the implications of this, and if there is a need for space to do so, maybe some of the descriptions of microstructures where they are similar in different taxa could be condensed.

There could maybe be more made of the unique position of the pulp cavity in phoebodonts. Are these the most basal chondrichthyes to have one? If so what are the implications for evolution of this- did it start near the tip of the tooth and move down to the top of the osteodentine later?

The retention of the early teeth on post functional positions is, as said, clearly holocephalan-like. This was discussed a bit in Johanson et al 2021 "The stem-holocephalan *Helodus* (Chondrichthyes; Holocephali) and the evolution of modern chimaeroid dentitions" where it is also noted that some other basal members of the holocephalan clade have teeth that are basally fused; this does not seem to be the case in the species figured here but does seem to be present in *Doliodus*. It would be useful if there was a discussion of the timing, and potentially mechanisms, for change between fused and unfused teeth, and whither there was a reversal of state and/or multiple events of teeth fusion.

***We have added a few sentences that incorporate the paper by Johanson et al 2021 and discuss their findings.

I assume you are using the term "elasmobranch" for the entire "shark clade" (ie closer to say *Carcharias* and *Raja* than *Chimaera*), as opposed to Maisey's restricted use. Maybe you should make this clear.

***We have changed large parts of the manuscript according to Reviewer 2's comments to give a better phylogenetic framework. In that sense, it is also clearer now what we refer to when using terms like elasmobranchs etc.